# Entanglement negativity and defect extremal surface

**Yilu Shao[1], Ma-Ke Yuan[1] and Yang Zhou[1,2]⋆**

**1** Department of Physics and Center for Field Theory and Particle Physics,
Fudan University, Shanghai 200433, China
**2** Peng Huanwu Center for Fundamental Theory, Hefei, Anhui 230026, China

⋆ yang_zhou@fudan.edu.cn

## Abstract

We study entanglement negativity for evaporating black hole based on the holographic model with defect brane. We introduce a defect extremal surface formula for entanglement negativity. Based on partial reduction, we show the equivalence between defect extremal surface formula and island formula for entanglement negativity in $AdS_3/BCFT_2$. Extending the study to the model of eternal black hole plus CFT bath, we find that black hole-black hole negativity decreases until vanishing, left black hole-left radiation negativity is always a constant, radiation-radiation negativity increases and then saturates at a time later than Page time. In all the time dependent cases, defect extremal surface formula agrees with island formula.

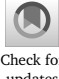

# 1  Introduction

Significant progress has been made in recent understanding of black hole information paradox [1–3]. In particular the island formula for the radiation gives Page curve [28–30] and therefore maintains unitarity. The development relies on the quantum extremal surface formula (QES formula) for the fine grained entropy, which was inspired from the quantum corrected Ryu-Takayanagi formula (RT formula) in computing holographic entanglement entropy [33–35]. While most of the recent studies have been centered on the von Neumann entropy, we need more detailed information about the quantum state, such as more general entanglement structures, to fully understand the black hole information problem. von Neu-

mann entropy is a unique measure characterizing entanglement between two subsystems $A$ and $B$ for a pure state $\psi_{AB}$. In this paper we want to study the entanglement between two subsystems in a mixed state. In particular we study entanglement negativity in an evaporating black hole with its Hawking radiation. There are several motivations to do so: First, the entire state of black hole and radiation is not always pure. Second, understanding the rich entanglement structures between subsystems of radiation is probably the key to understand how information escapes from the black hole.

As the analogy of von Neumann entropy for pure states, entanglement negativity is an important measure of entanglement in generally mixed states [4, 5]. The calculation of entanglement negativity in conformal field theories has been developed via the replica trick in [6–9]. The behavior of entanglement negativity was analysed in quantum many body systems [10–20, 55, 56] and in topological field theories [21–25]. The entanglement negativity in large central charge limit (large $c$ limit) was explored in [51]. Several attempts have been made to understand the holographic dual of entanglement negativity. In AdS$_3$/CFT$_2$, the lesson is that for two adjacent intervals, the holographic dual of entanglement negativity is given by the entanglement wedge cross section (times a constant factor).[1] Based on this, the quantum corrected holographic entanglement negativity and the island formula can be conjectured straightforwardly following the generalizations of holographic entanglement entropy [38, 40].

In this paper we propose defect extremal surface formula (DES formula) for entanglement negativity in holographic models with defects. Defect extremal surface is defined by extremizing the RT formula corrected by the quantum defect theory [46]. This is interesting when the AdS bulk contains a defect brane or string. The DES formula for entanglement negativity is a mixed state generalization of that for entanglement entropy. Based on a decomposition procedure of an AdS bulk with a brane, we demonstrate in this paper the equivalence between DES formula and island formula for negativity in AdS$_3$/BCFT$_2$. We also compute the evolution of entanglement negativity in evaporating black hole model and find that DES formula agrees with island formula.

**Note added.**  While this paper is in completion, we get to know the preprint [72] in arXiv, which has some overlap with this paper.

## 2   Review of entanglement negativity in CFT$_2$

Entanglement negativity, or more precisely logarithmic negativity, is a mixed state entanglement measure derived from the positive partial transpose criterion for the separability of mixed states. It can be defined as taking the trace norm of the partial transposed density matrix.

For a bipartite system, the partial transpose $\rho_{AB}^{T}$ of a density matrix $\rho_{AB}$ is defined by transposing only one part of the system, namely

$$\left\langle i_A, j_B \left| \rho_{AB}^{T_B} \right| k_A, l_B \right\rangle = \left\langle i_A, l_B \left| \rho_{AB} \right| k_A, j_B \right\rangle, \tag{1}$$

where $i_A, j_B, k_A, l_B$ are bases of $\mathcal{H}_{A,B}$ (the Hilbert space of subsystems $A, B$), and the entanglement negativity is then defined as

$$\mathcal{E}(A:B) = \mathcal{E}(\rho_{AB}) \equiv \log \left| \rho_{AB}^{T_B} \right|_1, \tag{2}$$

where $|\mathcal{O}|_1 = \text{Tr}\sqrt{\mathcal{O}\mathcal{O}^\dagger}$ is the trace norm.

---

[1]There is also an alternative proposal given by mutual information times a constant factor [62].

## 2.1 Replica trick for entanglement negativity

It is possible to calculate entanglement negativity in $(1+1)$d quantum field theories in analogy with the entanglement entropy by the replica trick.

The trace norm of partial transposed density matrix can be written in terms of its eigenvalues

$$\text{Tr}\left|\rho^{T_B}\right| = 1 + 2\sum_{\lambda_i < 0} |\lambda_i|. \tag{3}$$

Thus the integer power of the partial transposed density matrix

$$\text{Tr}\left(\rho^{T_B}\right)^n = \sum_i \lambda_i^n, \tag{4}$$

depends on the parity of $n$. Denoting $n_e = 2m$ and $n_o = 2m+1$ for some integer $m$, the analytic continuation with $n_e$ and $n_o$ will lead to different results. If we take $n_e \to 1$, we get our desired result of $\text{Tr}\left|\rho^{T_B}\right|$. If we take $n_o \to 1$, it only recovers the normalization $\text{Tr}\,\rho^{T_B} = 1$. This means that the correct way to perform analytic continuation is to consider the even sequence $n_e \to 1$, i.e.

$$\mathcal{E} = \lim_{n_e \to 1} \log \text{Tr}\left(\rho^{T_B}\right)^{n_e}. \tag{5}$$

To compute (5), we can use the replica trick introduced in [6]. Consider the system $AB = A \cup B$ made of two disjoint intervals $[u_1, v_1] \cup [u_2, v_2]$. Sewing $n$ copies of the original system along the branch cut representing subsystems $A$ and $B$ forms a $n$-sheet Riemann surface. The trace of the $n$-th power of the density matrix $\text{Tr}\,\rho_{AB}^n$ is the partition function $Z_n/(Z_1)^n$ on this $n$-sheet Riemann surface, with $Z_1$ the partition function of one copy of the original system.

The trace of the $n$-th power of the density matrix can also be written in terms of branch point twist fields as

$$\text{Tr}\,\rho_{AB}^n = \left\langle \mathcal{T}_n(u_1)\bar{\mathcal{T}}_n(v_1)\mathcal{T}_n(u_2)\bar{\mathcal{T}}_n(v_2)\right\rangle, \tag{6}$$

where $\mathcal{T}_n$ and $\bar{\mathcal{T}}_n$ are branch point twist fields with different boundary conditions.

Taking partial transpose of the density matrix $\rho_{AB}$ with respect to the second interval $B$ corresponds to the exchange of row and column indices in $B$. In the path integral representation, this is equivalent to interchanging the upper and lower edges of the second branch cut in $\rho_{AB}$. This interchange can be regarded as reversing the order of the column and row indices in the subsystem $B$.

Therefore, $\text{Tr}(\rho_{AB}^{T_B})^n$ is the partition function on the $n$-sheeted surface obtained by joining cyclically $n$ copies of subsystem $A$ and anti-cyclically of subsystem $B$. And the $n$-th power of the partial transposed density matrix can be written as

$$\text{Tr}\left(\rho_{AB}^{T_B}\right)^n = \left\langle \mathcal{T}_n(u_1)\bar{\mathcal{T}}_n(v_1)\bar{\mathcal{T}}_n(u_2)\mathcal{T}_n(v_2)\right\rangle. \tag{7}$$

We note here that throughout this paper, any $n$ appearing in the calculation of entanglement negativity using replica trick shall be automatically regarded as $n_e$.

## 2.2 Examples of entanglement negativity in CFT$_2$

In this subsection we review the results of some examples in CFT. It is known that the conformal weight of twist fields is

$$h_{\mathcal{T}_n} = h_{\bar{\mathcal{T}}_n} \equiv h_n = \frac{c}{24}\left(n - \frac{1}{n}\right), \tag{8}$$

where $c$ is the central charge of the CFT.

**Single interval.** We start with the four-point function in (7). Let $v_1 \to u_2$ and $v_2 \to u_1$, we have

$$\text{Tr}\left(\rho_{AB}^{T_B}\right)^n = \left\langle \mathcal{T}_n^2(u_2)\bar{\mathcal{T}}_n^2(v_2)\right\rangle. \tag{9}$$

Setting $n = n_e$, we get

$$\text{Tr}\left(\rho_{AB}^{T_B}\right)^{n_e} = \left(\left\langle \mathcal{T}_{n_e/2}(u_2)\bar{\mathcal{T}}_{n_e/2}(v_2)\right\rangle\right)^2 = d_{n_e/2}^2\left(\frac{u_2-u_1}{\epsilon}\right)^{-\frac{c}{3}\left(\frac{n_e}{2}-\frac{2}{n_e}\right)}, \tag{10}$$

with $\epsilon$ a UV regulator. $d_n$ is the OPE constant for the two-point function. And finally taking analytic continuation $n_e \to 1$ leads to

$$\left.\left|\rho_{AB}^{T_B}\right|\right|_1 = \lim_{n_e \to 1}\text{Tr}\left(\rho_{AB}^{T_B}\right)^{n_e} = d_{1/2}^2\left(\frac{\ell}{\epsilon}\right)^{c/2} \Rightarrow \mathcal{E} = \frac{c}{2}\log\left(\frac{\ell}{\epsilon}\right) + 2\log d_{1/2}, \tag{11}$$

where $\ell = u_2 - u_1$ denotes the length of the interval. From [6] we know that for pure state, entanglement negativity equals to Rényi entanglement entropy $S^{(n)}$ of order $1/2$. The latter is given by

$$S^{(n)} = \frac{c}{6}\left(1 + \frac{1}{n}\right)\log\left(\frac{\ell}{\epsilon}\right). \tag{12}$$

With $n \to 1/2$,

$$S^{(1/2)} = \frac{c}{2}\log\left(\frac{\ell}{\epsilon}\right). \tag{13}$$

Seen from holography [52], the minimal entanglement wedge cross section in this case is the RT surface. We can check that

$$\mathcal{E} = \frac{3}{2}E_W. \tag{14}$$

**Two adjacent intervals.** Let $v_1 \to u_2$ in (7), we have

$$\text{Tr}\left(\rho_{AB}^{T_B}\right)^n = \left\langle \mathcal{T}_n(u_1)\bar{\mathcal{T}}_n^2(u_2)\mathcal{T}_n(v_2)\right\rangle. \tag{15}$$

To keep it simple, set $u_1 = -\ell_1$, $u_2 = 0$ and $v_2 = \ell_2$ and all length are measured in the unit of $\epsilon$. The conformal dimension of the double twist operator $\mathcal{T}_n^2$ and $\bar{\mathcal{T}}_n^2$ is

$$h_{\mathcal{T}_n^2} = h_{\bar{\mathcal{T}}_n^2} \equiv h'_n = \frac{c}{12}\left(\frac{n}{2} - \frac{2}{n}\right). \tag{16}$$

Taking $n = n_e$ in (15), we get

$$\left\langle \mathcal{T}_{n_e}(-\ell_1)\bar{\mathcal{T}}_{n_e}^2(0)\mathcal{T}_{n_e}(\ell_2)\right\rangle = d_{n_e}^2\frac{C_{\mathcal{T}_{n_e}\bar{\mathcal{T}}_{n_e}^2\mathcal{T}_{n_e}}}{\ell_1^{2h'_{n_e}}\ell_2^{2h'_{n_e}}(\ell_1+\ell_2)^{4h_{n_e}-2h'_{n_e}}}. \tag{17}$$

The OPE structure constant $C_{\mathcal{T}_{n_e}\bar{\mathcal{T}}_{n_e}^2\mathcal{T}_{n_e}}$ is universal. We can actually fix this constant to be

$$\lim_{n_e \to 1} C_{\mathcal{T}_{n_e}\bar{\mathcal{T}}_{n_e}^2\mathcal{T}_{n_e}} = 2^{c/4}, \tag{18}$$

by comparing the Rényi reflected entropy and the entanglement negativity, see appendix A.[2] Taking $n_e \to 1$ and choose the normalization $d_1 = 1$, we have

$$\left.\left|\rho_A^{T_B}\right|\right|_1 \propto \left(\frac{\ell_1\ell_2}{\ell_1+\ell_2}\right)^{c/4} \cdot 2^{c/4} \Rightarrow \mathcal{E} = \frac{c}{4}\log\frac{\ell_1\ell_2}{(\ell_1+\ell_2)\epsilon} + \frac{c}{4}\log 2. \tag{19}$$

---

[2]This is an assumption that the entanglement negativity and half the $n = 1/2$ Rényi entropy coincide in the large $c$ limit, which is also the starting point of this paper. We will discuss this in detail in sec.3.1.

Recall the entanglement wedge cross section [41]

$$E_W = \begin{cases} \frac{c}{6} \log \frac{1+\sqrt{x}}{1-\sqrt{x}}, & \frac{1}{2} \leq x \leq 1, \\ 0, & 0 \leq x \leq \frac{1}{2}, \end{cases}$$

(20)

with $x$ the cross-ratio

$$x = \frac{\ell_1 \ell_2}{(\ell_1 + d)(\ell_2 + d)},$$

(21)

in which $d$ is the distance between two intervals. To recover adjacent interval limit, one can take $d = 2\epsilon \to 0$,

$$E_W \to \frac{c}{6} \log \left( \frac{2}{\epsilon} \frac{\ell_1 \ell_2}{(\ell_1 + \ell_2)} \right).$$

(22)

This again supports the relation

$$\mathcal{E} = \frac{3}{2} E_W.$$

(23)

For the more general two disjoint intervals the entanglement negativity depends on four-point function of a CFT, and it is challenging to compute and in most cases non-universal. Numerical methods are required. In [6], the asymptotic behavior of entanglement negativity of two disjoint intervals is studied. In the limit of $x \to 1$, the entanglement negativity is $\mathcal{E} \simeq -\frac{c}{4} \log(1-x)$. Following the monodromy method of Hartman [39], the authors of [51] recovers this result in the large $c$ limit.

## 3 Holographic results and island formula

### 3.1 Holographic computation of entanglement negativity

Kudler-Flam and Ryu proposed that the holographic dual of CFT logarithmic negativity $\mathcal{E}(A : B)$ is proportional to the entanglement wedge cross section in the classical gravity limit of AdS/CFT [52, 53]. They provided a derivation of the holographic dual of logarithmic negativity based on the observation that $n = 1/2$ Rényi reflected entropy[3] and entanglement negativity may coincide in the large $c$ limit of CFT$_2$

$$\mathcal{E} = \frac{1}{2} S_R^{(1/2)},$$

(24)

where $S_R^{(n)}$ is the Rényi reflected entropy of index $n$. It is therefore conjectured that the negativity has a holographic dual which is proportional to the area of the wedge cross section $\Gamma$ in the dual AdS space

$$\mathcal{E} = \frac{3}{2} E_W = \frac{3}{2} \frac{\text{Area}[\Gamma]}{4 G_N}.$$

(25)

Several checks have been done to support the conjectured formula (25):

- In [52], the results from holographic calculations agree with the CFT results derived by monodromy method in [51] near $x \sim 1$.

- In [53], the four-point function of twist operators are calculated using the Zamolodchikov recursion relation numerically and the authors find that the negativity matches the entanglement wedge cross section with a high precision.

---

[3]In a recent paper [75], the authors found counterexample (given by a special quantum state) where the reflected entropy is not a correlation measure. However, reflected entropy is still a valid correlation measure for holographic states [76]. See also [77] for the monotonicity property of reflected entropy in free fields.

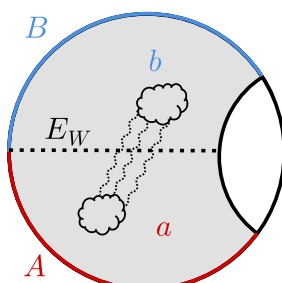

Figure 1: Schematic picture of quantum corrections to entanglement negativity (26). The quantum corrections come from bulk matters.

However, as pointed out by Dong, Qi and Walter in [54], the above derivation assumes the dominance of replica symmetric saddle. In general the replica non-symmetric saddle could dominate, which may take the holographic dual of logarithmic negativity for two disjoint intervals away from the wedge cross section. In this paper we will restrict ourselves to adjacent two intervals.

Kudler-Flam and Ryu also conjectured the quantum corrected logarithmic negativity formula

$$\mathcal{E}(A:B) = \frac{3}{2} \frac{\langle \mathcal{A}[\partial a \cap \partial b] \rangle_{\tilde{\rho}_{ab}}}{4G_N} + \mathcal{E}^{\text{bulk}}(a:b) + \mathcal{O}(G_N), \tag{26}$$

where the entanglement wedge of $AB$ is divided into two regions $a,b$ by the cross section $\partial a \cap \partial b$, and $\mathcal{A}$ is the area operator.[4] $\mathcal{E}^{\text{bulk}}(a:b)$ is the logarithmic negativity for the density matrix $\tilde{\rho}_{ab}$ of the bulk field theory, as shown in fig.1.

The quantum corrected logarithmic negativity formula (26) is similar to the Faulkner, Lewkowycz and Maldacena (FLM) formula of entanglement entropy [37]. Notice that FLM formula only computes the first two orders as an approximation. Engelhardt and Wall proposed that holographic entanglement entropy can be calculated exactly [38] in bulk Plank constant using the so called QES formula which extremizes the generalized entropy (which coincides with FLM if evaluated on the classical minimal surface).[5] In the same spirit, it is tempting to conjecture a quantum extremal cross section which can provide exact result for logarithmic negativity. This leads us to the QES formula for logarithmic negativity

$$\mathcal{E}(A:B) = \text{ext}_{Q'} \left\{ \frac{3}{2} \frac{\text{Area}(Q' = \partial a \cap \partial b)}{4G_N} + \mathcal{E}^{\text{bulk}}(a:b) \right\}, \tag{27}$$

where the quantum extremal cross section is denoted by $Q'$. We emphasize that $A$ and $B$ are adjacent intervals.

## 3.2 Island formula for entanglement negativity

It has been found recently that QES formula can be generalized to island formula for von Neumann entropy. See [32] for a review. Given that there is a QES formula for logarithmic negativity, it is tempting to generalize it to gravitational system. In later sections we will discuss explicitly the two-dimensional eternal black hole + CFT model of black hole evaporation, where the generalizations of QES formula for logarithmic negativity can be justified. There a black hole version of the generalized QES formula can be easily written down

$$\mathcal{E}(B_L:B_R) = \min \text{ext}_{Q'} \left\{ \frac{3}{2} \frac{\text{Area}(Q' = \partial b_L \cap \partial b_R)}{4G_N} + \mathcal{E}(\tilde{\rho}_{b_L}:\tilde{\rho}_{b_R}) \right\}, \tag{28}$$

---

[4]Here we focus on the static case and employ quantum extremal surface (instead of RT surface of $AB$) to define the entanglement wedge of $AB$.

[5]See [43] for further discussions.

where $b_L$ and $b_R$ are the entanglement wedges for left black hole and right black hole respectively. Accordingly the island formula of logarithmic negativity for radiation is[6]

$$\mathcal{E}(A:B) = \min \operatorname{ext}_{Q'} \left\{ \frac{3}{2} \frac{\operatorname{Area}\left(Q' = \partial \operatorname{Is}(A) \cap \partial \operatorname{Is}(B)\right)}{4G_N} + \mathcal{E}\left(A \cup \operatorname{Is}(A) : B \cup \operatorname{Is}(B)\right) \right\}. \quad (29)$$

We emphasize again that $A$ and $B$ are adjacent intervals. In the remaining text, we refer to the first term in $\{\cdots\}$ of (28) or (29) as "area term" and the second term as "matter term". We call the entirety in $\{\cdots\}$ "the generalized (entanglement) negativity". We leave more detail discussions about island formula of logarithmic negativity to sec.6 and sec.7. We also note that in [71], the island formula is obtained by considering the Rényi reflected entropy in large $c$ limit through (24).[7]

## 4 Holographic BCFT model with bulk defect

In [42], Takayanagi proposed a holographic dual for $BCFT_2$ by considering a classical $AdS_3$ bulk truncated by a boundary codimension-one brane $Q$ with Neumann boundary condition imposed on it. The bulk action can be written as

$$I = \frac{1}{16\pi G_N} \int_B \sqrt{-g}(R - 2\Lambda) + \frac{1}{8\pi G_N} \int_Q \sqrt{-h}(K - T), \quad (30)$$

where $B$ and $Q$ stand for the bulk and the brane respectively, and $T$ is the constant brane tension. By variation of the bulk action, we get the Neumann boundary condition on the brane

$$K_{ab} = (K - T)h_{ab}, \quad (31)$$

where $h_{ab}$ is the induced metric and $K_{ab}$ the extrinsic curvature of the brane. The $AdS_3$ bulk metric can be written as

$$\begin{aligned} ds^2 &= d\rho^2 + l^2 \cosh^2 \frac{\rho}{l} \cdot \frac{-dt^2 + dy^2}{y^2} \\ &= \frac{l^2}{z^2}\left(-dt^2 + dx^2 + dz^2\right), \end{aligned} \quad (32)$$

with $l$ the $AdS_3$ radius, and the relation between the coordinates $(\rho, y)$ and $(x, z)$ is as follow

$$z = -y / \cosh \frac{\rho}{l}, \qquad x = y \tanh \frac{\rho}{l}. \quad (33)$$

Assume that the brane $Q$ is stationary at a constant position $\rho = \rho_0 > 0$, where the constant $\rho_0$ is related to $T$ by [46–48]

$$T = \frac{\tanh\left(\frac{\rho_0}{l}\right)}{l}. \quad (34)$$

In the remaining part of this paper, we also use the polar coordinate $\theta$, which is related to $\rho$ via $\frac{1}{\cos\theta} = \cosh\left(\frac{\rho}{l}\right)$, thus the brane is located at

$$\theta_0 = \arccos\left[\cosh\left(\frac{\rho_0}{l}\right)\right]^{-1} > 0. \quad (35)$$

---

[6]In the context of AdS/CFT [52, 53], the prefactor 3/2 in (25) is true only for 3-dimensional bulk geometry. We will see that the coefficient is still preserved in the 2d effective theory description following partial Randall-Sundrum reduction. The detail will be discussed in sec.4.

[7]See also related works [59–70, 72–74].

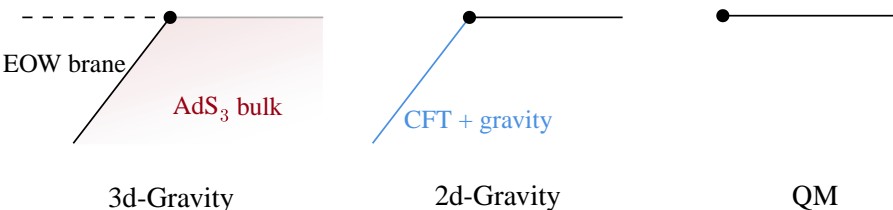

Figure 2: Three different descriptions of the holographic BCFT model.

The boundary entropy can be evaluated from the disk partition function. The difference of the partition function between $\rho = 0$ and $\rho = \rho_0$ is given by $I_E(\rho_0) - I_E(0) = -\frac{\rho_0}{4G_N}$. Then we obtain the boundary entropy

$$S_{\text{bdy}} = \frac{\rho_0}{4G_N}. \tag{36}$$

Now we introduce the Holographic BCFT model with conformal matter on the brane. This model is inspired by the work of Almheiri, Mahajan, Maldacena and Zhao [29] but we treat the theory on the brane differently. Instead of replacing the brane matter by a part of AdS wedge, we treat the conformal matter as defect theory on the brane embedded in the bulk. Moreover, we obtain the 2d gravity on the brane from the partial Randall-Sundrum (R-S) reduction of the bulk.

Similiar to [29], our model has three alternative descriptions as illustrated in fig.2:

- 3d-Gravity: 3d gravity theory in $AdS_3$ with an End-Of-the-World (EOW) brane as a bulk defect on part of the space ($x < 0$), and with a rigid AdS boundary on the rest ($x > 0$). There is conformal matter on the EOW brane.

- 2d-Gravity: 2d CFT + gravity theory living on $x < 0$ coupled to a 2d CFT living on $x > 0$. The gravity is obtained by partial R-S reduction.

- QM: A two-dimensional CFT on the half-line $x > 0$ with particular boundary degrees of freedom at $x = 0$. This description should be viewed as the fundamental one.

Now we describe how to change from the 3d-gravity description to the 2d-gravity description via partial R-S reduction and AdS/CFT correspondence.[8] As illustrated in fig.3, starting with the 3d-gravity description, we first decompose the $AdS_3$ bulk into $W_1$ and $W_2$, where $W_2$ is half the entire $AdS_3$ space. For $W_1$, we perform the brane world reduction, or the Randall-Sundrum reduction [45], along the extra dimension $\rho$ to obtain a 2d gravity theory on the EOW brane $Q$. For $W_2$, we replace it with the half-space CFT according to AdS/CFT correspondence. Finally we get the 2d effective theory which is a brane gravity theory with CFT on it glued to a flat half-space CFT. Note that during this process, as shown in fig.3, the part of the geodesic in $W_1$ (i.e. the red arc in fig.3), whose length is $\text{arctanh}(\sin\theta_0)$, is reduced to the area term in the island formula (29) of the 2d effective theory description, and this explains the 3/2 coefficient of the area term in the island formula (29).

The effective Newton constant on the EOW brane is

$$\frac{1}{4G_N^{(2)}} = \frac{\rho_0}{4G_N} = \frac{c}{6}\text{arctanh}(\sin\theta_0). \tag{37}$$

This is interpreted as boundary entropy (36) in the original AdS/BCFT proposal [42].

---

[8]Details of this reduction can be found in [46].

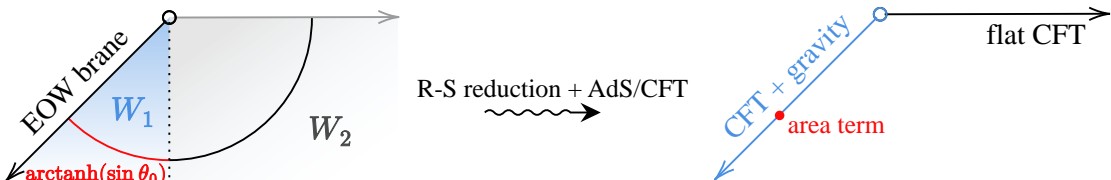

Figure 3: (Modified from [46]) The reduction procedure, during which $W_1$ is reduced to the gravity on the EOW brane and $W_2$ is dual to a flat half-space CFT. The part of the geodesic in $W_1$ (the red arc) is reduced to the area term in the 2d effective theory description.

**Discussion: Karch-Randall brane world model.** Here we add the discussion on the relationship between our model and the model proposed in [78,79], where the AdS bulk bounded by Karch-Randall brane is expected to be dual to two CFTs and the one on the brane may be called as an inherent CFT. This is not our perspective in this paper. By partial reduction we perform explicit dimension reduction (which may not be applicable in higher dimensions) for the AdS$_3$ gravity action between Karch-Randall brane and the tensionless brane. The resulting 2d gravity is therefore equivalent to 3d gravity in that region, which means that we do not need additional duality to translate this part AdS gravity to some inherent CFT. In our set up we treat the brane CFT as a bulk defect representing some bulk degrees of freedom from the beginning. Our perspective has received a bunch of tests [46–48].

## 5  Entanglement negativity on bulk defect

Now we return to the 3d-gravity description and calculate the entanglement negativity on bulk defect. If the tension of the EOW brane $Q$ is zero, the EOW brane will be orthogonal to the asymptotic boundary. By adding matter or turn on the tension in the viewpoint of [42], the EOW brane can move to a position with constant angle $\theta_0$. According to [30], the CFT on AdS$_2$ can be mapped to a BCFT in flat space via a Weyl transformation. The Weyl factor can be read from the induced metric on the brane $ds^2_{\text{brane}} = \Omega^{-2}(y)ds^2_{\text{flat}}$, i.e.

$$\Omega(y) = \left| \frac{y \cos \theta_0}{l} \right| . \tag{38}$$

### 5.1  Single interval $[0, y]$ on the brane

From [8], the calculation of a single interval $[0, y]$ including the boundary point means considering the one-point function of the double twist operator $\mathcal{T}_n^2$ inserted at $y$. The conformal invariance fixes the form of one-point function on a flat BCFT and by the analysis of twist operators in [8,44],[9]

$$\left\langle \mathcal{T}_n^2(y) \right\rangle_{\text{flat}} = \left\langle \mathcal{T}_{n/2}(y) \right\rangle^2_{\text{flat}} = \frac{g_n}{|2y/\epsilon_y|^{2h'_n}} , \tag{39}$$

with $\epsilon_y$ the UV cut-off on the brane.

From Weyl transformation (38), the one-point function on the brane is

$$\left\langle \mathcal{T}_n^2(y) \right\rangle_Q = \left| \frac{y \cos \theta_0}{l} \right|^{2h'_n} \left\langle \mathcal{T}_n^2(y) \right\rangle_{\text{flat}} = g_n \left| \frac{\epsilon_y \cos \theta_0}{2l} \right|^{2h'_n} . \tag{40}$$

---

[9]From now on we will use $\langle \cdots \rangle_{\text{flat}}$ for correlation function on flat BCFT, $\langle \cdots \rangle_Q$ for correlation function on brane, and $\langle \cdots \rangle$ for correlation function on flat CFT.

Finally, the entanglement negativity on the brane is obtained by taking $n \to 1$ of the brane one-point function

$$\begin{aligned} \mathcal{E}_{\text{defect}} &= \lim_{n \to 1} \log \langle \mathcal{T}_n^2(y) \rangle_Q \\ &= \frac{c}{4} \log \frac{2l}{\epsilon_y \cos \theta_0} + \log g \,. \end{aligned} \tag{41}$$

In our model, the boundary do not admit physical degrees of freedom, so we can pick $\log g = 0$. Notice that in this case the entanglement negativity on the brane is a constant and does not depend on the length of the interval.

## 5.2 Single interval $[y_1, y_2]$ on the brane

Now we derive the the entanglement negativity on the brane for a single interval $[y_1, y_2]$. Following [8] we insert two double twist operators $\mathcal{T}_n^2$ and $\bar{\mathcal{T}}_n^2$ at $y = y_1$ and $y = y_2$ respectively. Applying Weyl transformation, the two-point function is given by

$$\langle \mathcal{T}_n^2(y_1) \bar{\mathcal{T}}_n^2(y_2) \rangle_Q = \left| \frac{y_1 \cos \theta_0}{l} \right|^{2h_n'} \left| \frac{y_2 \cos \theta_0}{l} \right|^{2h_n'} \langle \mathcal{T}_n^2(y_1) \bar{\mathcal{T}}_n^2(y_2) \rangle_{\text{flat}} \,. \tag{42}$$

Using the doubling trick, this BCFT two-point function can be seen as a four-point function in the full plane. Employing the same trick developed in [44] for BCFT entanglement entropy, one can find analytical results in the large $c$ limit.

Inspired by the holographic calculation [39, 40, 44], we expect that the BCFT two-point function has two possible dominate channels: the *operator product expansion channel* (OPE) and the *boundary operator expansion channel* (BOE). This corresponds to different way of doing operator product expansion, see [44]. The dominant channel can be determined by the cross-ratio

$$\eta(y_1, y_2) = \frac{4y_1 y_2}{(y_1 - y_2)^2} \,. \tag{43}$$

From the holographic side, this two channel endures a phase transition due to the change of RT surface from the connected phase to the disconnected phase.

**OPE channel.** If $\eta \to \infty$, the OPE channel dominates. The corresponding two point function is

$$\langle \mathcal{T}_n^2(y_1) \bar{\mathcal{T}}_n^2(y_2) \rangle_{\text{flat}} = \frac{\epsilon_y^{4h_n'}}{(y_1 - y_2)^{4h_n'}} \,. \tag{44}$$

The entanglement negativity is derived from (42):

$$\mathcal{E}_{\text{defect}} = \frac{c}{4} \log \frac{l^2(y_1 - y_2)^2}{y_1 y_2 \epsilon_y^2 \cos^2 \theta_0} \,. \tag{45}$$

**BOE channel.** If $\eta \to 0$, the BOE channel dominates. The corresponding two point function is

$$\langle \mathcal{T}_n^2(y_1) \bar{\mathcal{T}}_n^2(y_2) \rangle_{\text{flat}} = \frac{g_b^{4(1-n/2)} \epsilon_y^{4h_n'}}{(4y_1 y_2)^{2h_n'}} \,. \tag{46}$$

The entanglement negativity is then:

$$\mathcal{E}_{\text{defect}} = \frac{c}{2} \log \frac{2l}{\epsilon_y \cos \theta_0} \,. \tag{47}$$

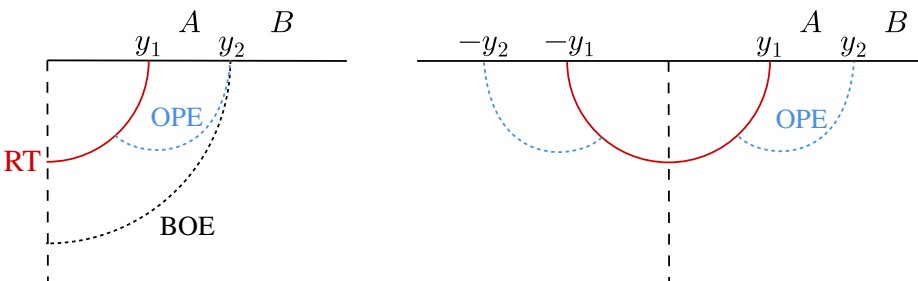

Figure 4: (Modified from [48]) Left: The holographic dual of possible channels. The boundary point of the brane BCFT is dual to an EOW brane denoted by the dashed vertical line. The RT surface for $A \cup B$ is denoted by the red arc. From the field theory point of view, the BOE channel corresponds to a product of two one-point BCFT correlators. Right: The OPE channel corresponds to a four-point function on a whole CFT, and the four-point function duals to the cross section (blue dashed arcs).

By the same analysis in [46], in the large $c$ limit, OPE channel dominates when $\eta \to \infty$, otherwise the BOE channel dominates. Equating these two results, we have the phase transition point at $\eta = 1$.

## 5.3 Adjacent intervals $[y_1, y_2]$ and $[y_2, \infty]$ on the brane

For adjacent intervals on brane, we need to consider a different boundary two-point function. We insert $\mathcal{T}_n$ and $\bar{\mathcal{T}}_n^2$ at $y = y_1$ and $y = y_2$ respectively. The two-point function would be

$$\left\langle \mathcal{T}_n(y_1) \bar{\mathcal{T}}_n^2(y_2) \right\rangle_Q = \left| \frac{y_1 \cos \theta_0}{l} \right|^{2h_n} \left| \frac{y_2 \cos \theta_0}{l} \right|^{2h_n'} \left\langle \mathcal{T}_n(y_1) \bar{\mathcal{T}}_n^2(y_2) \right\rangle_{\text{flat}} . \tag{48}$$

**BOE channel.** By the same analysis as above (also see fig.4) we break the two-point function into product of two one-point function, so the method in [44] still works. The result reads

$$\left\langle \mathcal{T}_n(y_1) \bar{\mathcal{T}}_n^2(y_2) \right\rangle_{\text{flat}} = \frac{g_b^{(3-2n)} \epsilon_y^{2h_n + 2h_n'}}{(2y_1)^{2h_n} (2y_2)^{2h_n'}} , \tag{49}$$

thus the entanglement negativity is

$$\mathcal{E}_{\text{defect}} = \frac{c}{4} \log \frac{2l}{\epsilon_y \cos \theta_0} . \tag{50}$$

**OPE channel.** For the simple case in which there is no degree of freedom on the boundary, we can evaluate the analytic results by making use of the doubling trick. The doubling trick maps a BCFT to a chiral CFT on the flat plane. The BCFT two-point function is now mapped to a four-point function in the chiral CFT, which is in general hard to compute. However, if we assume large $c$ limit and vacuum block dominance, this four-point function can be calculated numerically and one can check that the dominate channels are indeed the ones corresponding to the holographic configurations, as illustrated in fig.4. In Appendix B we give a numerical

check and show that[10]

$$\lim_{n \to 1} \left\langle \mathcal{T}_n(y_1) \bar{\mathcal{T}}_n^2(y_2) \right\rangle_{\text{flat}} = \lim_{n \to 1} \left\langle \mathcal{T}_n'^2(-y_2) \bar{\mathcal{T}}_n'(-y_1) \mathcal{T}_n'(y_1) \bar{\mathcal{T}}_n'^2(y_2) \right\rangle$$

$$= 2^{(c/2)/2} \left[ \frac{(y_2 + y_1)(y_2 - y_1)}{(2y_1)\epsilon_y} \right]^{c/4} . \tag{51}$$

The entanglement negativity on bulk defect is then obtained,

$$\mathcal{E}_{\text{defect}} = \frac{c}{4} \log \frac{2l}{\xi \epsilon_y \cos \theta_0} , \tag{52}$$

with

$$\xi = \frac{2y_2 y_1}{y_2^2 - y_1^2} . \tag{53}$$

Note that the $\xi$ defined here is slightly different from the ordinary cross-ratio. At large $c$ limit, the OPE channel dominates in the case of $\xi > 1$, while the BOE channel dominates in the case of $\xi < 1$. The critical value $\xi_c = 1$ is given by equating (50) and (52).

## 5.4 Adjacent intervals $[0, y_2]$ and $[y_2, y_3]$ on the brane

Following the previous twist operator calculation, we insert $\bar{\mathcal{T}}_n^2$ and $\mathcal{T}_n$ at $y = y_2$ and $y = y_3$ respectively. The two-point function would be:

$$\left\langle \bar{\mathcal{T}}_n^2(y_2) \mathcal{T}_n(y_3) \right\rangle_Q = \left| \frac{y_2 \cos \theta_0}{l} \right|^{2h_n'} \left| \frac{y_3 \cos \theta_0}{l} \right|^{2h_n} \left\langle \bar{\mathcal{T}}_n^2(y_2) \mathcal{T}_n(y_3) \right\rangle_{\text{flat}} . \tag{54}$$

**BOE channel.** The two-point function is given by

$$\left\langle \bar{\mathcal{T}}_n^2(y_2) \mathcal{T}_n(y_3) \right\rangle_{\text{flat}} = \frac{g_b^{(3-2n)} \epsilon_y^{2h_n'+2h_n}}{(2y_2)^{2h_n'}(2y_3)^{2h_n}} , \tag{55}$$

as above, thus the entanglement negativity for BOE channel is

$$\mathcal{E}_{\text{defect}} = \frac{c}{4} \log \frac{2l}{\epsilon_y \cos \theta_0} . \tag{56}$$

**OPE channel.** In this channel the entanglement negativity is

$$\mathcal{E}_{\text{defect}} = \frac{c}{4} \log \frac{2l}{\xi \epsilon_y \cos \theta_0} , \tag{57}$$

with

$$\xi = \frac{2y_3 y_2}{y_3^2 - y_2^2} , \tag{58}$$

and the phase transition point is also at $\xi_c = 1$.

## 6 Defect extremal surface for entanglement negativity

In this section, we will propose the defect extremal surface formula for entanglement negativity and compare the results from DES and island formula in single interval and adjacent intervals.

---

[10]In (51) we use $\mathcal{T}'$ and $\bar{\mathcal{T}}'$ to represent the chiral operators, which have half the conformal weight of the original operators.

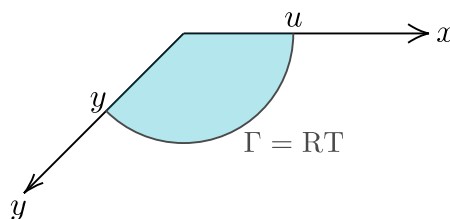

Figure 5: The case of a single interval $[0, u]$. The blue area denotes the entanglement wedge of this single interval.

## 6.1 DES: The proposal

In the 3d-gravity description, we consider a quantum theory living on the defect which is regarded as a part of the full bulk theory since it is coupled to the bulk. When the classical RT surface terminates on the defect, the defect theory should contribute to the entanglement negativity. Following the idea of [46] we propose the *defect extremal surface* formula for entanglement negativity including defect contribution[11]

$$\mathcal{E}_{\text{DES}} = \min_{\Gamma, X} \left\{ \text{ext}_{\Gamma, X} \left[ \frac{3}{2} \frac{\text{Area}(\Gamma)}{4 G_N} + \mathcal{E}_{\text{defect}}[D] \right] \right\}, \quad X = \Gamma \cap D, \tag{59}$$

where $\Gamma$ is a $1d$ curve in AdS$_3$, $D$ denotes the defect, and $X$ is the $0d$ entangling surface given by the intersection of $\Gamma$ and $D$. $\mathcal{E}_{\text{defect}}$ is the entanglement negativity on bulk defect, which is derived in the previous section. We will call the $[\cdots]$ part of (59) "the generalized negativity",[12] with its first term "the area term" and the second term "the defect term".

## 6.2 Single interval $[0, u]$

### 6.2.1 DES result

In this case we consider a single interval which contains the boundary point. The curve $\Gamma$ can only end on the brane, as shown in fig.5. Assuming that the intersection point $X$ is located at $y$, then the length of $\Gamma$ can be derived from the same geometric analysis in [46]. We denote

$$u' = \frac{y^2 + u^2 + 2yu \sin \theta_0}{2(u + y \sin \theta_0)}, \tag{60}$$

$$\theta_0' = \arcsin \frac{u^2 + 2yu \sin \theta_0 - y^2 \cos 2\theta_0}{u^2 + 2yu \sin \theta_0 + y^2}. \tag{61}$$

The generalized negativity is given by considering both contributions in (59)

$$\mathcal{E}_{\text{gen}}(y) = \mathcal{E}_{\text{area}} + \mathcal{E}_{\text{defect}} = \frac{c}{4} \log \frac{2u'}{\epsilon_u} + \frac{c}{4} \text{arctanh}(\sin \theta_0') + \frac{c'}{4} \log \frac{2l}{\epsilon_y \cos \theta_0}, \tag{62}$$

where $c'$ denotes the central charge of the defect CFT, and we use the single interval bulk defect result (41).

---

[11]Our proposal eq.(59) only works for matter localized at EOW brane, and should be improved for the cases that the bulk matter is distributed in a more general fashion. In the DES formula, both extremizations over the local shape of RT surface and the location of the end point are involved. The geodesic connecting two given points is unique.

[12]Since we will always distinguish the calculation from DES and island formula, the generalized negativity here will not be confused with the generalized negativity in (28) or (29), and the same is true for the area term below.

In order to extremize $\mathcal{E}_{\text{gen}}(y)$, let $\partial_y \mathcal{E}_{\text{gen}}(y) = 0$, the intersection point of defect extremal surface and the brane is located at

$$y = u, \tag{63}$$

which means that the extremal surface is the same as the RT surface. This is expected because $\mathcal{E}_{\text{defect}}$ coming from brane matter is a constant. The resulting entanglement negativity from DES is

$$\mathcal{E}_{\text{DES}} = \frac{c}{4} \log \frac{2u}{\epsilon_u} + \frac{c}{4} \operatorname{arctanh}(\sin \theta_0) + \frac{c'}{4} \log \frac{2l}{\epsilon_y \cos \theta_0}. \tag{64}$$

### 6.2.2 Island result

Here we switch to the 2d-gravity description. Since we have already proposed the island formula of the entanglement negativity, i.e. (28) and (29), we will calculate it explicitly below.

In this case, it can be viewed as a single interval with length $u + y$. From the calculation in [46], we can see that

$$\mathcal{E}_{\text{eff}}(x_1, x_2) = \frac{c}{4} \log \left( \frac{|x_1 - x_2|^2}{\epsilon_1 \epsilon_2 \Omega(x_1, \bar{x}_1) \Omega(x_2, \bar{x}_2)} \right), \tag{65}$$

where $\epsilon_{1,2}$ are the UV cut-offs and $\Omega$ comes from the Weyl factor of the metric $\mathrm{d}s^2 = \Omega^{-2} \mathrm{d}x \mathrm{d}\bar{x}$. Thus we derive the matter term

$$\mathcal{E}_{\text{eff}} = \frac{c}{4} \log \frac{(u+y)^2 l}{\epsilon_u \epsilon_y y \cos \theta_0}. \tag{66}$$

The generalized negativity is obtained by adding the area term

$$\begin{aligned} \mathcal{E}_{\text{gen}}(y) &= \mathcal{E}_{\text{eff}}([-y, u]) + \mathcal{E}_{\text{area}}(y) \\ &= \frac{c}{4} \log \frac{(u+y)^2 l}{\epsilon_u \epsilon_y y \cos \theta_0} + \frac{c}{4} \operatorname{arctanh}(\sin \theta_0). \end{aligned} \tag{67}$$

Extremizing (67) gives the location of the quantum extremal surface

$$y = u, \tag{68}$$

which is the same as the end point of the defect extremal surface. The entanglement negativity is

$$\begin{aligned} \mathcal{E}_{\text{QES}} &= \frac{c}{4} \log \frac{4ul}{\epsilon_u \epsilon_y \cos \theta_0} + \frac{c}{4} \operatorname{arctanh}(\sin \theta_0) \\ &= \frac{c}{4} \log \frac{2u}{\epsilon_u} + \frac{c}{4} \operatorname{arctanh}(\sin \theta_0) + \frac{c}{4} \log \frac{2l}{\epsilon_y \cos \theta_0}. \end{aligned} \tag{69}$$

If we consider the simple case that $c' = c$, this result would recover our previous result (64). The bulk DES result agrees with the QES result from island formula.

## 6.3 Single interval $[u_1, u_2]$

### 6.3.1 DES result

Here we consider a general single interval which does not contain the boundary point. In this case the entanglement negativity has a phase transition due to the variation of the cross-ratio between the operators. Note that the "phase" below always refers to the phase of the BCFT on the rigid AdS boundary.

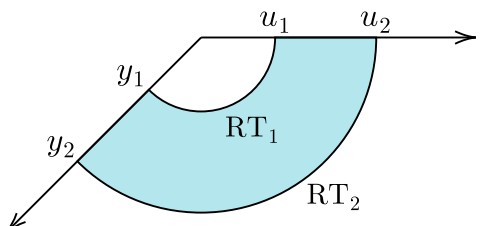

Figure 6: The disconnected phase of a single interval $[u_1, u_2]$. Disconnected phase means that the RT surface intersects with the brane.

**Phase-I: Connected phase.** In this phase, the extremal surface does not intersect with the brane. Thus the defect term $\mathcal{E}_{\text{defect}}$ vanishes and the generalized entanglement negativity is given by the boundary contribution only, i.e.

$$\mathcal{E}_{\text{gen}} = \frac{c}{2} \log \frac{(u_2 - u_1)}{\epsilon_u}, \tag{70}$$

which is a trivial single-interval result. See [8] for exact calculation.

**Phase-II: Disconnected phase.** The defect extremal surface terminates on the brane as shown in fig.6 and the entanglement negativity of an interval $[-y_1, -y_2]$ on the brane contributes. When the cross-ratio on the brane $\eta \to \infty$, the generalized negativity is given by.

$$\begin{aligned}
\mathcal{E}_{\text{gen}}(y_1, y_2) &= \mathcal{E}_{\Gamma_1} + \mathcal{E}_{\Gamma_2} + \mathcal{E}_{\text{defect}} \\
&= \frac{c}{4} \Bigg( \log \frac{y_1^2 + u_1^2 + 2y_1 u_1 \sin\theta_0}{(u_1 + y_1 \sin\theta_0)\epsilon_u} + \text{arctanh} \frac{u_1^2 + 2y_1 u_1 \sin\theta_0 - y_1^2 \cos 2\theta_0}{u_1^2 + 2y_1 u_1 \sin\theta_0 + y_1^2} \\
&\quad + \log \frac{y_2^2 + u_2^2 + 2y_2 u_2 \sin\theta_0}{(u_2 + y_2 \sin\theta_0)\epsilon_u} + \text{arctanh} \frac{u_2^2 + 2y_2 u_2 \sin\theta_0 - y_2^2 \cos 2\theta_0}{u_2^2 + 2y_2 u_2 \sin\theta_0 + y_2^2} \\
&\quad + \log \frac{l^2 (y_1 - y_2)^2}{y_1 y_2 \epsilon_y^2 \cos^2 \theta_0} \Bigg).
\end{aligned} \tag{71}$$

By extremizing $\mathcal{E}_{\text{gen}}(y_1, y_2)$ with respect to $y_1$ and $y_2$, we find that $\partial_y \mathcal{E}_{\text{gen}}(y_1, y_2) < 0$ for any $y_1$ and $y_2$ thus there is no extremal surface. When $\eta \to 0$,

$$\begin{aligned}
\mathcal{E}_{\text{gen}}(y_1, y_2) &= \mathcal{E}_{\Gamma_1} + \mathcal{E}_{\Gamma_2} + \mathcal{E}_{\text{defect}} \\
&= \frac{c}{4} \Bigg( \log \frac{y_1^2 + u_1^2 + 2y_1 u_1 \sin\theta_0}{(u_1 + y_1 \sin\theta_0)\epsilon_u} + \text{arctanh} \frac{u_1^2 + 2y_1 u_1 \sin\theta_0 - y_1^2 \cos 2\theta_0}{u_1^2 + 2y_1 u_1 \sin\theta_0 + y_1^2} \\
&\quad + \log \frac{y_2^2 + u_2^2 + 2y_2 u_2 \sin\theta_0}{(u_2 + y_2 \sin\theta_0)\epsilon_u} + \text{arctanh} \frac{u_2^2 + 2y_2 u_2 \sin\theta_0 - y_2^2 \cos 2\theta_0}{u_2^2 + 2y_2 u_2 \sin\theta_0 + y_2^2} \\
&\quad + 2 \log \frac{2l}{\epsilon_y \cos\theta_0} \Bigg).
\end{aligned} \tag{72}$$

By extremizing $\mathcal{E}_{\text{gen}}(y_1, y_2)$ with respect to $y_1$ and $y_2$, i.e. $\partial_{y_1} \mathcal{E}_{\text{gen}}(y_1, y_2) = \partial_{y_2} \mathcal{E}_{\text{gen}}(y_1, y_2) = 0$, we get the location of the intersection of defect extremal surface and the EOW brane

$$y_1 = u_1, \qquad y_2 = u_2, \tag{73}$$

which means that $\Gamma$ is the same as the RT surface. Following DES proposal we obtain

$$\mathcal{E}_{\text{DES}} = \frac{c}{4} \left( \log \frac{2u_1}{\epsilon_u} + \log \frac{2u_2}{\epsilon_u} + 2\,\text{arctanh}(\sin\theta_0) + 2 \log \frac{2l}{\epsilon_y \cos\theta_0} \right). \tag{74}$$

To summarize, the final results are

$$
\mathcal{E}_{\mathrm{DES}} = \begin{cases} \frac{c}{2} \log \frac{(u_1 - u_2)}{\epsilon_u}, & \eta \to \infty, \\ \frac{c}{4} \left[ \log \frac{4u_1 u_2}{\epsilon_u^2} + 2 \operatorname{arctanh}(\sin\theta_0) + 2 \log \frac{2l}{\epsilon_y \cos\theta_0} \right], & \eta \to 0. \end{cases}
\tag{75}
$$

### 6.3.2 Island result

**Phase-I: Connected phase.** In this case, the negativity only includes the matter term.

$$
\mathcal{E}_{\mathrm{QES}} = \frac{c}{2} \log \frac{(u_2 - u_1)}{\epsilon}.
\tag{76}
$$

**Phase-II: Disconnected phase.** Since the brane CFT is coupled to gravity, we should consider the contribution of the interval $[-y_1, -y_2]$ on the brane. The end points of the interval have corresponding area term which is given by

$$
\mathcal{E}_{\mathrm{area}} = 2 \times \frac{1}{4G_N^{(2)}} = \frac{c}{2} \operatorname{arctanh}(\sin\theta_0).
\tag{77}
$$

Considering the derivation of the negativity of multiple intervals at large $c$ in [39, 51], the generalized negativity is given by

$$
\begin{aligned}
\mathcal{E}_{\mathrm{gen}}(a, b) &= \mathcal{E}_{\mathrm{area}} + \mathcal{E}_{\mathrm{eff}}\big([-y_1, -y_2] \cup [u_1, u_2]\big) \\
&= \frac{c}{2} \operatorname{arctanh}(\sin\theta_0) \\
&\quad + \min\left\{ \frac{c}{4} \log \frac{(y_1 - y_2)^2 (u_1 - u_2)^2 l^2}{y_1 y_2 \epsilon_u^2 \epsilon_y^2 \cos^2\theta_0}, \frac{c}{4} \log \frac{(u_1 + y_1)^2 (u_2 + y_2)^2 l^2}{y_1 y_2 \epsilon_u^2 \epsilon_y^2 \cos^2\theta_0} \right\},
\end{aligned}
\tag{78}
$$

where the two terms in {} correspond to the $\eta \to \infty$ and $\eta \to 0$ case respectively, with $\eta$ the cross-ratio defined in (43). We note that the general analytic behavior of entanglement negativity in this case is very different from entanglement entropy [51], the critical point of $\eta$ (the phase-transition point) should be determined by numerical calculation, but it will not affect our current discussion. As we have assumed the large $c$ limit, from holographic side we can determine the critical point $\eta_c$ by equating these two results.

In the minimization procedure, we can see for the first term $\partial_{y_2} \mathcal{E}_{\mathrm{gen}}(y_1, y_2) < 0$, which means that there is no extremal point. Taking extremization of the second term gives

$$
y_1 = u_1, \qquad y_2 = u_2.
\tag{79}
$$

Thus the final negativity is given by

$$
\mathcal{E}_{\mathrm{QES}} = \frac{c}{2} \operatorname{arctanh}(\sin\theta_0) + \frac{c}{4} \log \frac{16 u_1 u_2 l^2}{\epsilon_u^2 \epsilon_y^2 \cos^2\theta_0}.
\tag{80}
$$

To summarize,

$$
\mathcal{E}_{\mathrm{QES}} = \begin{cases} \frac{c}{2} \log \frac{(u_2 - u_1)}{\epsilon_u}, & \eta > \eta_c, \\ \frac{c}{4} \left[ \log \frac{4u_1 u_2}{\epsilon_u^2} + 2 \operatorname{arctanh}(\sin\theta_0) + 2 \log \frac{2l}{\epsilon_y \cos\theta_0} \right], & \eta < \eta_c, \end{cases}
\tag{81}
$$

which is exactly the same as (75).

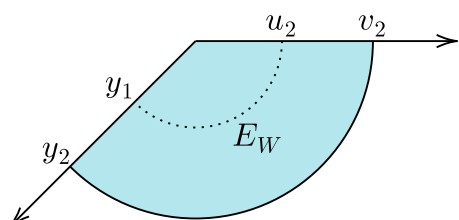

Figure 7: The case of adjacent intervals $[0, u_2]$ and $[u_2, v_2]$.

## 6.4   Adjacent intervals $[0, u_2]$ and $[u_2, v_2]$

### 6.4.1   DES result

In this case we consider adjacent intervals on the boundary including the boundary point of BCFT, as illustrated in fig.7. Let $\Gamma$ be the minimal surface in AdS$_3$ with two endpoints $u_2$ and $y_1$, thus the area term is given by

$$\mathcal{E}_{\text{area}} = \frac{3}{2} \frac{\text{Area}(\Gamma)}{4G_N} = \frac{3L(y_1)}{8G_N} . \tag{82}$$

Now we combine the result with the calculation results from sec.5.4. For $\xi < 1$ the defect contribution is constant. Extremization of $\mathcal{E}_{\text{gen}}(y_1)$ over $y_1$ gives the location of the intersection point of defect extremal surface and the brane:

$$y_1 = u_2 , \tag{83}$$

which leads to

$$\mathcal{E}_{\text{DES}} = \text{ext}_{y_1}\{\mathcal{E}_{\text{area}} + \mathcal{E}_{\text{defect}}\} = \frac{c}{4} \log \frac{2u_2}{\epsilon_u} + \frac{c}{4} \text{arctanh}(\sin\theta_0) + \frac{c}{4} \log \frac{2l}{\epsilon_y \cos\theta_0} . \tag{84}$$

For $\xi > 1$, we have

$$\mathcal{E}_{\text{gen}} = \frac{3L(y_1)}{8G_N} + \frac{c}{4} \log \frac{2l}{\xi \epsilon_y \cos\theta_0} . \tag{85}$$

For (85), we found $\partial_{y_1}\mathcal{E}_{\text{gen}} < 0$. So there are no extremal surface and brane contributions.

### 6.4.2   Island result

Note that in large $c$ limit, the four-point function factorizes. The RT surface in the outer side serves actually as an IR cut-off so it does not contribute to the negativity. When $\xi < 1$, the four-point function factorizes as

$$\left\langle \mathcal{T}_n(y_2) \bar{\mathcal{T}}_n^2(y_1) \mathcal{T}_n^2(u_2) \bar{\mathcal{T}}_n(v_2) \right\rangle = \left\langle \bar{\mathcal{T}}_n^2(y_1) \mathcal{T}_n^2(u_2) \right\rangle \left\langle \mathcal{T}_n(y_2) \bar{\mathcal{T}}_n(v_2) \right\rangle , \tag{86}$$

which leads to

$$\mathcal{E}_{\text{eff}} = \frac{c}{4} \log \frac{(u_2 + y_1)^2 l}{\epsilon_u \epsilon_y y_1 \cos\theta_0} . \tag{87}$$

Adding the area term and doing extremization over $y_1$ gives $u_2 = y_1$, as we expected. The entanglement negativity is

$$\mathcal{E}_{\text{DES}} = \text{ext}_{y_1}\{\mathcal{E}_{\text{area}} + \mathcal{E}_{\text{eff}}\} = \frac{c}{4} \log \frac{2u_2}{\epsilon_u} + \frac{c}{4} \text{arctanh}(\sin\theta_0) + \frac{c}{4} \log \frac{2l}{\epsilon_y \cos\theta_0} . \tag{88}$$

We can see precise agreement between (88) and (84).

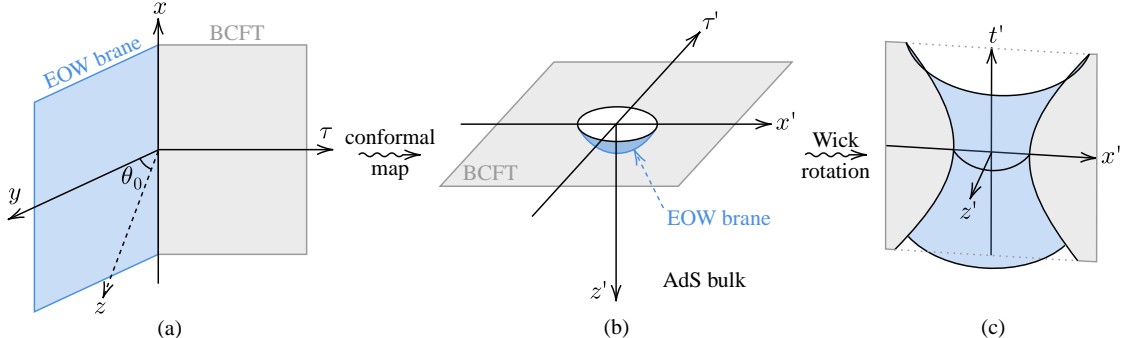

Figure 8: (Modified from [47]) (a) The bulk description of the Euclidean BCFT$_2$. (b) The geometry of our model after the conformal transformation (90), with the BCFT$_2$'s boundary mapped to a unit circle (91) and the EOW brane mapped to a spherical cap (92). (c) The Lorentz geometry of the AdS$_3$/BCFT$_2$ model.

## 7 Time dependent entanglement negativity in 2d eternal black hole

In this section we investigate the time dependent entanglement negativity in the 2d eternal black hole emerging from the boundary effective description of the AdS$_3$/BCFT$_2$ setting [47, 48]. In sec.6 we demonstrate the consistency between the entanglement negativity computed by the bulk DES formula and that computed by the boundary island formula for a static time slice. Now we further consider the time dependent case and show that this consistency still holds. We also calculate the time evaluation of the entanglement negativity between different parts of this black hole system.

### 7.1 Review of the 2d eternal black hole system

Firstly we look at the emergency of the 2d eternal black hole. In sec.4 we see that the holographic dual of BCFT$_2$ is an AdS$_3$ bulk bounded by its asymptotic boundary and an EOW brane. The Euclidean AdS$_3$ metric is given by

$$ds^2 = \frac{l^2}{z^2}\left(d\tau^2 + dx^2 + dz^2\right),\tag{89}$$

and the EOW brane is placed at plane $\tau = -z\tan\theta_0$ as depicted in fig.8(a). To clarify the physical interpretation, we perform the following conformal transformation

$$\begin{aligned}\tau &= \frac{2(x'^2 + \tau'^2 + z'^2 - 1)}{(\tau'+1)^2 + x'^2 + z'^2},\\ x &= \frac{4x'}{(\tau'+1)^2 + x'^2 + z'^2},\\ z &= \frac{4z'}{(\tau'+1)^2 + x'^2 + z'^2}.\end{aligned}\tag{90}$$

As shown in fig.8(b), the conformal transformation (90) maps the BCFT$_2$'s boundary ($\tau = z = 0$) to a unit circle

$$x'^2 + \tau'^2 = 1,\tag{91}$$

and the EOW brane to a spherical cap

$$(z' + \tan\theta_0)^2 + x'^2 + \tau'^2 = \sec^2\theta_0,\tag{92}$$

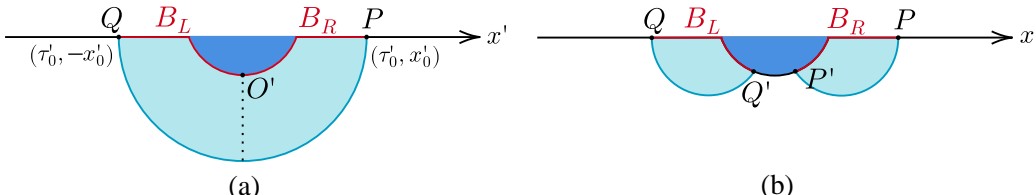

Figure 9: (a) The connected phase of the extremal surface for the black hole. (b) The disconnected phase. In this phase an island ($Q'P'$) appearing in the middle separating the whole black hole into two parts. From now on we will use solid blue lines to represent the extremal surfaces, light blue regions for entanglement wedges, dark blue regions for EOW branes, black dashed lines for entanglement cross sections $E_W$, solid red lines for black hole regions and solid black lines for islands.

while preserving the metric form

$$ds^2 = \frac{l^2}{z'^2}\left(d\tau'^2 + dx'^2 + dz'^2\right) . \tag{93}$$

Employing the decomposition reviewed in sec.4 gives the 2d effective boundary description in which the EOW brane is a gravitational region and it is surrounded by a bath CFT as depicted in fig.8(b). To see where is the black hole, one can analytical continue the Euclidean time to Lorentz time ($\tau' \to it'$) as shown in fig.8(c). In Lorentz spacetime, the boundary of the EOW brane (91) becomes $x'^2 - t'^2 = 1$. We then introduce the Rindler coordinates ($T, X$)

$$x' = e^X \cosh T , \qquad t' = e^X \sinh T . \tag{94}$$

In this coordinate system the metric takes the form as Rindler space thus describes the near-horizon geometry of a black hole [47].

## 7.2 The entanglement negativity between black hole interiors

In this section we study the entanglement negativity between black hole interiors, i.e. $B_L$ and $B_R$ shown in fig.9. Following the setting in [47, 48], the black hole is identified as the space-like interval with $Q(t'_0, -x'_0)$ and $P(t'_0, x'_0)$ as its two endpoints. We will first perform our calculation in Euclidean coordinates ($\tau, x, z$) or ($\tau', x', z'$), then analytically continue to Lorentz coordinates by $\tau' \to it'$ and finally use (94) to get the time evolution.

### 7.2.1 Bulk description

**Phase-I: Connected phase.** To calculate $\mathcal{E}(B_L : B_R)$ from the bulk point of view we first need to determine the entanglement wedge of $B_L \cup B_R$. As shown in fig.9(a), the entanglement wedge of the whole black hole is the light blue region bounded by a space-like interval $QP$ on the boundary and the extremal surface which is a geodesic connecting $Q$ and $P$ in the AdS$_3$ bulk. The cross section connects the extremal surface (the blue arc in fig.9(a)) and the EOW brane (the dark blue region). To employ the DES formula (59), one has to combine the area term from the cross section with the defect term from the defect contribution and then do extremization. The defect term in (59) is given by the single interval result (41), i.e.

$$\mathcal{E}_{\text{defect}}(B_L : B_R) = \frac{c}{4} \log \frac{2l}{\epsilon_y \cos\theta_0} , \tag{95}$$

which is a constant thus the extremization only needs to be performed on the area term. The calculation of the minimal cross section is similar to eq.(5.20) in [48], which gives

$$\text{ext}_\Gamma \, \text{Area}(\Gamma) = l \left[ \log \frac{x_0'^2 + \tau_0'^2 - 1 + \sqrt{4x_0'^2 + (\tau_0'^2 + x_0'^2 - 1)^2}}{2x_0'} + \log \frac{\cos\theta_0}{1 - \sin\theta_0} \right]. \tag{96}$$

Finally the BH-BH entanglement negativity in the connected phase is given by

$$\mathcal{E}_{\text{conn}}^{\text{bulk}}(B_L : B_R) = \frac{c}{4} \left[ \log \frac{x_0'^2 + \tau_0'^2 - 1 + \sqrt{4x_0'^2 + (\tau_0'^2 + x_0'^2 - 1)^2}}{2x_0'} \right.$$
$$\left. + \log \frac{\cos\theta_0}{1 - \sin\theta_0} + \log \frac{2l}{\epsilon_y \cos\theta_0} \right]. \tag{97}$$

**Phase-II: Disconnected phase.** As shown in fig.9(b), in the disconnected phase the extremal surfaces intersect the EOW brane at $Q'$ and $P'$ and the region $Q'P'$ corresponds to an island. The island splits the black hole thus the entanglement wedge of $B_L$ and $B_R$ are separated, resulting in a vanishing area term. The defect term is given by

$$\mathcal{E}_{\text{defect}} = \lim_{n \to 1} \log \Omega_{Q'}^{2h_n} \Omega_{P'}^{2h_n} \left\langle \bar{\mathcal{T}}_n(Q') \bar{\mathcal{T}}_n(P') \right\rangle_{\text{flat}}$$
$$= \lim_{n \to 1} \left| \frac{y_{Q'} \cos\theta_0}{l} \right|^{2h_n} \left| \frac{y_{P'} \cos\theta_0}{l} \right|^{2h_n} \log \left\langle \bar{\mathcal{T}}_n(Q') \bar{\mathcal{T}}_n(P') \right\rangle_{\text{flat}}. \tag{98}$$

By using the doubling trick one can express the two-point correlation function on BCFT as a chiral CFT four-point correlation function on the whole plane

$$\left\langle \bar{\mathcal{T}}_n(Q') \bar{\mathcal{T}}_n(P') \right\rangle_{\text{flat}} = \left\langle \bar{\mathcal{T}}_n'(Q') \bar{\mathcal{T}}_n'(P') \mathcal{T}_n'(P'') \mathcal{T}_n'(Q'') \right\rangle, \tag{99}$$

where $P''(y = -\tau_0, x = x_0)$ and $Q''(-\tau_0, -x_0)$ are the mirror images of $P'$ and $Q'$ with respect to the plane $\tau = 0$. Assuming the large $c$ limit, then the correlator is factorized into contractions

$$\left\langle \bar{\mathcal{T}}_n'(Q') \bar{\mathcal{T}}_n'(P') \mathcal{T}_n'(P'') \mathcal{T}_n'(Q'') \right\rangle = \left\langle \bar{\mathcal{T}}_n'(P') \mathcal{T}_n'(P'') \right\rangle \left\langle \bar{\mathcal{T}}_n'(Q') \mathcal{T}_n'(Q'') \right\rangle. \tag{100}$$

The two-point function is given by (note that the power is halved due to the chiral operators)

$$\left\langle \bar{\mathcal{T}}_n'(P') \mathcal{T}_n'(P'') \right\rangle = \lim_{n \to 1} d_n |P'P''|^{-4h_n/2} = 1, \tag{101}$$

thus the defect term vanishes in this case. Finally the BH-BH entanglement negativity in the disconnected phase is given by

$$\mathcal{E}_{\text{disconn}}^{\text{bulk}}(B_L : B_R) = 0. \tag{102}$$

### 7.2.2 Boundary description

**Phase-I: Connected phase.** To employ the boundary island formula we first calculate the matter term, which is given by the three-point correlator

$$\mathcal{E}_{\text{eff}}(B_L : B_R) = \lim_{n \to 1} \log \Omega_{O'}^{2h_n'} \left\langle \mathcal{T}_n(Q) \bar{\mathcal{T}}_n^2(O') \mathcal{T}_n(P) \right\rangle$$
$$= \lim_{n \to 1} \log \Omega_{O'}^{2h_n'} C_{\mathcal{T}\bar{\mathcal{T}}^2\mathcal{T}}^n |O'Q|^{-2h_n'} |O'P|^{-2h_n'} |QP|^{2h_n' - 4h_n} \epsilon_y^{2h_n'}. \tag{103}$$

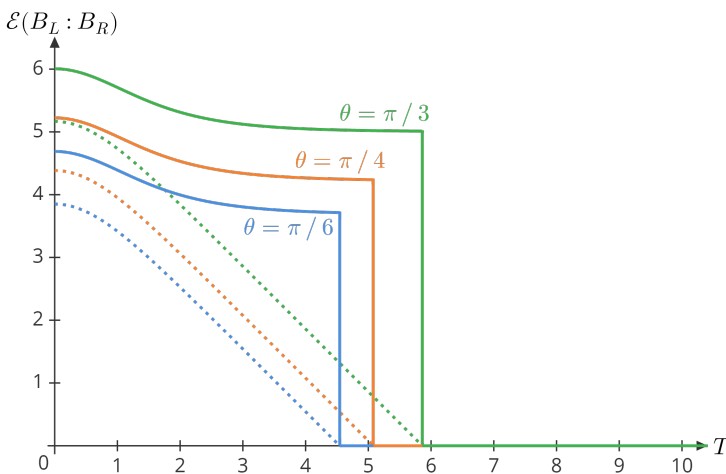

Figure 10: The entanglement negativity between black hole interiors (in the unit of $\frac{c}{4}$) with respect to time $T$ for $X_0 = 1$ and $\theta = \frac{\pi}{6}, \frac{\pi}{4}, \frac{\pi}{3}$. We pick $\epsilon_y = 0.1$ and $l = 1$. The dash lines refer to $\frac{3}{4}$ mutual information, which decreases to zero at the Page time.

Inserting $|O'Q| = |O'P| = \sqrt{(\tau_0 + y)^2 + x_0^2}$ and $|QP| = 2x_0$ into (103) and combining the area term gives

$$\mathcal{E}_{\text{gen}}^{\text{bdy}}(B_L : B_R) = \frac{c}{4}\left[\log\frac{l}{y\cos\theta_0} + \log 2 + \log\frac{x_0^2 + (\tau_0 + y)^2}{2x_0\epsilon_y} + \text{arctanh}(\sin\theta_0)\right]. \quad (104)$$

Extremizing (104) over the position $y$ gives $y = \sqrt{x_0^2 + \tau_0^2}$, thus the final result is

$$\mathcal{E}_{\text{conn}}^{\text{bdy}}(B_L : B_R) = \frac{c}{4}\left[\log\frac{\tau_0 + \sqrt{\tau_0^2 + x_0^2}}{x_0} + \log\frac{2l}{\epsilon_y\cos\theta_0} + \text{arctanh}(\sin\theta_0)\right], \quad (105)$$

which agrees with (97).

**Phase-II: Disconnected phase.** In the disconnected phase the two parts of black hole do not intersect as shown in fig.9(b), thus in the boundary calculation there is no area term contribution and we only need to consider the matter term, which is given by the four-point correlation function

$$\mathcal{E}_{\text{gen}}^{\text{bdy}}(B_L : B_R) = \mathcal{E}_{\text{eff}}(B_L : B_R) = \lim_{n\to 1}\log\Omega_{Q'}^{2h_n}\Omega_{P'}^{2h_n}\left\langle\mathcal{T}_n(Q)\bar{\mathcal{T}}_n(Q')\bar{\mathcal{T}}_n(P')\mathcal{T}_n(P)\right\rangle. \quad (106)$$

As implied by the disconnected extremal surface illustrated in fig.9(b), assuming large $c$ limit, this four-point correction function factorizes into two two-point functions

$$\left\langle\mathcal{T}_n(Q)\bar{\mathcal{T}}_n(Q')\bar{\mathcal{T}}_n(P')\mathcal{T}_n(P)\right\rangle = \left\langle\mathcal{T}_n(Q)\bar{\mathcal{T}}_n(Q')\right\rangle\left\langle\bar{\mathcal{T}}_n(P')\mathcal{T}_n(P)\right\rangle = 1. \quad (107)$$

Thus in this phase the boundary result is

$$\mathcal{E}_{\text{disconn}}^{\text{bdy}}(B_L : B_R) = 0, \quad (108)$$

which is consistent with the bulk calculation.

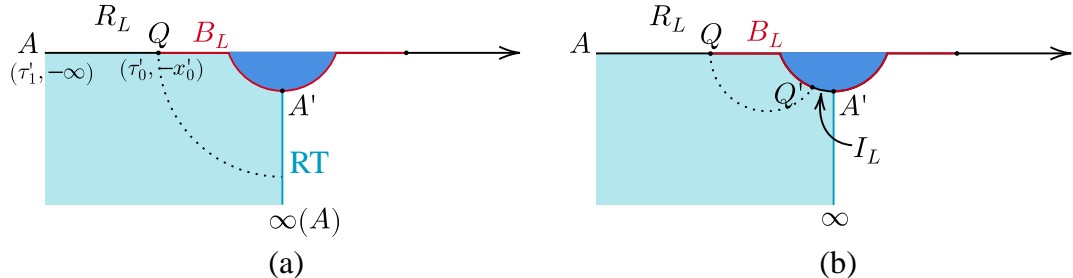

Figure 11: The RT surface connects $A$ at infinity and $A'$ on the brane. (a) The connected phase. (b) The disconnected phase.

### 7.2.3 Time evolution

Here we rewrite the results of $\mathcal{E}(B_L : B_R)$ in the Rindler coordinates $(T, X)$ using (94)

$$\mathcal{E}(B_L : B_R) = \begin{cases} \frac{c}{4}\left[\log\frac{e^{2X_0}-1+\sqrt{4e^{2X_0}\cosh^2 T + (e^{2X_0}-1)^2}}{2e^{X_0}\cosh T} + \log\frac{\cos\theta_0}{1-\sin\theta_0} + \log\frac{2l}{\epsilon_y\cos\theta_0}\right], & T < T_P, \\ 0, & T > T_P, \end{cases} \tag{109}$$

where $X_0$ is a fixed constant describing the black hole boundary and $T_P$ is the Page time given in [47]

$$T_P = \text{arccosh}\left(\sinh X_0 e^{\text{arctanh}(\sin\theta_0)}\frac{2l}{\epsilon_y\cos\theta_0}\right). \tag{110}$$

In fig.10 the time-dependent BH-BH entanglement negativity is plotted under specific parameters. The BH-BH entanglement negativity decreases at first and shifts to zero at Page time. Note that it follows the same curve of the BH-BH reflected entropy (fig.13 in [48]) up to a 3/4 factor.

## 7.3 The entanglement negativity between left radiation and left black hole

In this section we compute the entanglement negativity between left radiation and left black hole. Here the left part radiation refers to the interval $AQ$ with its two endpoints $A(\tau'_1, -\infty)$ and $Q(\tau'_0, -x'_0)$ in the $(\tau', x', z')$ coordinates as illustrated in fig.11. The boundary of the left part black hole is $Q$.

### 7.3.1 Bulk description

**Phase-I: Connected phase.** As shown in fig.11(a), the entanglement wedge of $R_L \cup B_L$ is bounded by the boundary and associated extremal surface, which is a geodesic connects $A$ to the brane. In this phase the entanglement wedge cross section does not intersect with the brane, thus $\mathcal{E}_{\text{defect}}$ vanishes and the generalized entanglement negativity is given by $\mathcal{E}_{\text{area}}$ only. The minimal cross section is given by (D.3) with $\tau'_0 = \tau'_1$, $\tau'_1 = \tau'_0$, $x'_0 = -x'_1 = -\infty$ and $x'_1 = -x'_0$. Finally the entanglement negativity between left radiation and left black hole in this phase is given by

$$\mathcal{E}_I^{\text{bulk}}(R_L : B_L)$$
$$= \lim_{x'_1 \to \infty} \frac{c}{4}\log\frac{2\sqrt{[(\tau'_1-\tau'_0)^2+(x'_1-x'_0)^2][(\tau'_1\tau'_0-1)^2+(x'_1x'_0-1)^2+\tau'^2_1 x'^2_0+\tau'^2_0 x'^2_1-1]}}{\epsilon(-1+\tau'^2_1+x'^2_1)}$$
$$= \frac{c}{4}\log\frac{2\sqrt{x'^2_0+\tau'^2_0}}{\epsilon}. \tag{111}$$

**Phase-II: Disconnected phase.** In this phase the cross section intersects with the EOW brane and an island appears as shown in fig.11(b). To calculate the entanglement negativity in this case we have to take into account the contribution from the defect, which is given by the adjacent intervals result (56)

$$\mathcal{E}_{\text{defect}}(I_L : B_L) = \frac{c}{4} \log \frac{2l}{\epsilon_y \cos\theta_0} \,, \tag{112}$$

at large $c$ limit. The calculation of the minimal cross section is similar to eq.(3.7) in [47]. Add them together and do extermization, and one gets the entanglement negativity between $R_L$ and $B_L$ in this phase

$$\begin{aligned}
\mathcal{E}_{\text{II}}^{\text{bulk}}(R_L : B_L) &= \min_{\Gamma, X} \left\{ \text{ext}_{\Gamma, X} \left[ \mathcal{E}_{\text{area}}(R_L \cup I_L : R_L) + \mathcal{E}_{\text{defect}}(I_L : B_L) \right] \right\} \\
&= \frac{c}{4} \left( \log \frac{x_0'^2 + \tau_0'^2 - 1}{\epsilon} + \text{arctanh}(\sin\theta_0) + \log \frac{2l}{\epsilon_y \cos\theta_0} \right).
\end{aligned} \tag{113}$$

### 7.3.2 Boundary description

**Phase-I: Connected phase.** The matter term in the connected phase is given by the entanglement negativity between the intervals $AQ$ and $QA'$, which can be computed by the three-point correlation function

$$\mathcal{E}_{\text{eff}}(R_L : B_L) = \lim_{n \to 1} \log \Omega_{A'}^{2h_n} \left\langle \mathcal{T}_n(A) \bar{\mathcal{T}}_n^2(Q) \mathcal{T}_n(A') \right\rangle. \tag{114}$$

We can calculate the three-point correlator in the $(\tau, x, z)$ coordinates, where we have $A(2, 0, 0)$, $Q(\tau_0, -x_0, 0)$ and $A'(-2\sin\theta_0, 0, 2\cos\theta_0)$. In this phase the cross section does not terminate on the brane thus there is no area term. Therefore, the negativity between $R_L$ and $B_L$ reads

$$\begin{aligned}
\mathcal{E}_{\text{I}}^{\text{bdy}}(R_L : B_L) &= \lim_{n \to 1} \log \Omega_{A'}^{2h_n} C_{\mathcal{T}\bar{\mathcal{T}}^2\mathcal{T}}^n |AQ|^{-2h_n'} |QA'|^{-2h_n'} |AA'|^{2h_n' - 4h_n} \tilde{\epsilon}^{2h_n'} \\
&= \frac{c}{4} \left[ \log 2 + \log \frac{\sqrt{(\tau_0 - 2)^2 + x_0^2} \sqrt{(\tau_0 + 2)^2 + x_0^2}}{4} - \log \frac{4\epsilon}{(\tau_0' + 1)^2 + x_0'^2} \right],
\end{aligned} \tag{115}$$

where $\tilde{\epsilon}$ is the $(\tau', x')$-dependent UV cut-off in the $(\tau, x, z)$ coordinates and it corresponds to the last term in the third line.[13] One can check that (115) coincides with the bulk result (111) exactly.

**Phase-II: Disconnected phase.** In the disconnected phase the left side radiation has an entanglement island $I_L = Q'A'$ as illustrated in fig.11(b), thus the matter term in island formula is the entanglement negativity between the left side radiation plus its island and the left side black hole, which reads

$$\mathcal{E}_{\text{eff}}(R_L \cup I_L : B_L) = \lim_{n \to 1} \log \Omega_{Q'}^{2h_n'} \Omega_{A'}^{2h_n} \left\langle \mathcal{T}_n(A) \bar{\mathcal{T}}_n^2(Q) \mathcal{T}_n^2(Q') \bar{\mathcal{T}}_n(A') \right\rangle. \tag{116}$$

At large $c$ limit, the four-point correlator factorizes into two two-point correlators

$$\left\langle \mathcal{T}_n(A) \bar{\mathcal{T}}_n^2(Q) \mathcal{T}_n^2(Q') \bar{\mathcal{T}}_n(A') \right\rangle = \left\langle \mathcal{T}_n(A) \bar{\mathcal{T}}_n(A') \right\rangle \left\langle \bar{\mathcal{T}}_n^2(Q) \mathcal{T}_n^2(Q') \right\rangle. \tag{117}$$

---

[13]In sec.7 we take the UV cut-off of the asymptotic boundary a constant in the $(\tau', x', z')$ coordinates, i.e. $z' = \epsilon$. Thus in the $(\tau, x, z)$ coordinates the cut-off is $(\tau', x')$-dependent.

We assume that in the $(\tau, x, z)$ coordinates $Q'$ is located at $(-y \sin \theta_0, x, y \cos \theta_0)$, then using (44) one obtain

$$\mathcal{E}_{\text{gen}}^{\text{bdy}}(R_L : B_L) = \frac{c}{4} \operatorname{arctanh}(\sin \theta_0) + \frac{c}{4} \log \frac{l}{y \cos \theta_0}$$
$$+ \frac{c}{4} \left[ \log \frac{\sqrt{(\tau_0 + y)^2 + (x_0 - x)^2}^2}{\epsilon_y} - \log \frac{4\epsilon}{(\tau_0' + 1)^2 + x_0'^2} \right]. \tag{118}$$

Extremizing the generalized entanglement negativity over $x$ and $y$ gives the position of $Q' : y = \tau_0$, $x = x_0$. Finally the Rad-BH entanglement negativity in the disconnected phase is given by

$$\mathcal{E}_{\text{II}}^{\text{bdy}}(R_L : B_L) = \frac{c}{4} \left[ \operatorname{arctanh}(\sin \theta_0) + \log \frac{l}{\tau_0 \cos \theta_0} + \log \frac{4\tau_0^2}{\epsilon_y} - \log \frac{4\epsilon}{(\tau_0' + 1)^2 + x_0'^2} \right]$$
$$= \frac{c}{4} \left[ \operatorname{arctanh}(\sin \theta_0) + \log \frac{2l}{\epsilon_y \cos \theta_0} + \log \frac{x_0'^2 + \tau_0'^2 - 1}{\epsilon} \right], \tag{119}$$

which agrees with (113) exactly.

### 7.3.3 Time evolution

Combining (111) and (113), one can rewrite the entanglement negativity between radiation and black hole in Rindler coordinates $(T, X)$ using (94)

$$\mathcal{E}_{\text{I}}(R_L : B_L) = \frac{c}{4} \log \frac{2e^{X_0}}{\epsilon},$$
$$\mathcal{E}_{\text{II}}(R_L : B_L) = \frac{c}{4} \left( \log \frac{e^{2X_0} - 1}{\epsilon} + \operatorname{arctanh}(\sin \theta_0) + \log \frac{2l}{\epsilon_y \cos \theta_0} \right). \tag{120}$$

Note that in both phases the negativity is a constant, thus the final result is given by

$$\mathcal{E}(R_L : B_L) = \min\{\mathcal{E}_{\text{I}}(R_L : B_L), \mathcal{E}_{\text{II}}(R_L : B_L)\}. \tag{121}$$

## 7.4 The entanglement negativity between radiation and radiation

In this section we consider the entanglement negativity between two adjacent regions of radiation, i.e. the nearby radiation $R_N$ and distant radiation $R_D$ as illustrated in fig.12 and fig.13. The nearby part radiation $R_N$ refers to the interval $MQ \cup PN$ with its four endpoints $M(-T + i\pi, X_1)$, $Q(-T + i\pi, X_0)$, $P(T, X_0)$ and $N(T, X_1)$ in the Rindler coordinates. While the distant part $R_D$ refers to $EM \cup NF$ with its four endpoints $E(-T + i\pi, X_2)$, $M$, $N$ and $F(T, X_2)$. The $(\tau', x', z')$ coordinates of the endpoints are shown in fig.12(a) and they can be mapped from the $(T, X)$ coordinates via

$$x_0' = e^{X_0} \cosh T, \qquad \tau_0' = ie^{X_0} \sinh T,$$
$$x_1' = e^{X_1} \cosh T, \qquad \tau_1' = ie^{X_1} \sinh T, \tag{122}$$
$$x_2' = e^{X_2} \cosh T, \qquad \tau_2' = ie^{X_2} \sinh T.$$

### 7.4.1 Bulk description

**Phase-I.** As shown in fig.12(a), in phase-I there is no defect term contribution so we only need to compute the cross section separating $R_N$ and $R_D$ (the dashed curve in fig.12(a)), thus we get

$$\mathcal{E}_{\text{I}}^{\text{bulk}}(R_N : R_D) = \frac{c}{2} \log \frac{2x_1'}{\epsilon}. \tag{123}$$

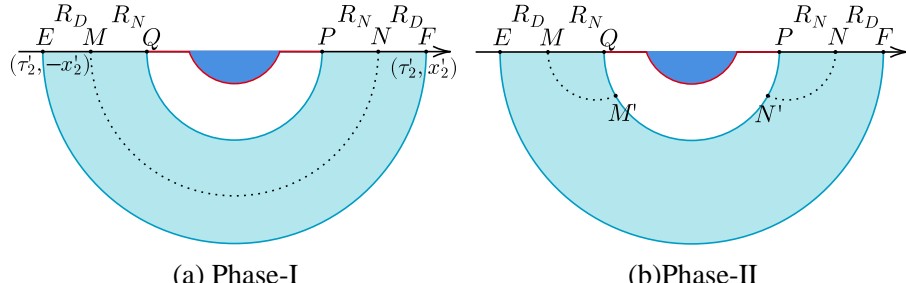

(a) Phase-I  (b)Phase-II

Figure 12: Possible configurations of the connected phase. (a) The cross section does not terminate on the extremal surface $\widehat{QP}$. (b) The cross sections terminate on $\widehat{QP}$. In this paper we do not consider the phases in which the cross section terminates on the outer extremal surface $\widehat{EF}$ and the phases in which $Q$ is connected to $E$ as one can make these phases never appear by taking the value of $X_2$ to be large enough.

**Phase-II.** In phase-II the defect term vanishes and we only need to calculate $E_W(R_N : R_D)$ in fig.12(b). The length of the geodesic connecting $N(\tau_1', x_1')$ and the extremal surface $\widehat{QP} : x'^2 + z'^2 = x_0'^2$ & $\tau' = \tau_0'$ is given in appendix C and we denote this length by $L_2$

$$L_2 = \log \frac{\sqrt{\left[(\tau_0' - \tau_1')^2 + x_0'^2 + x_1'^2\right]^2 - 4x_0'^2 x_1'^2}}{\epsilon x_0'} . \tag{124}$$

Then the Rad-Rad entanglement negativity in phase-II from the bulk description is given by

$$\mathcal{E}_{\text{II}}^{\text{bulk}}(R_N : R_D) = \frac{c}{2} \log \frac{\sqrt{\left[(\tau_0' - \tau_1')^2 + x_0'^2 + x_1'^2\right]^2 - 4x_0'^2 x_1'^2}}{\epsilon x_0'} . \tag{125}$$

**Phase-III.** As shown in fig.13(a), in phase-III an island appears. However, the entire island belongs to $R_N$ therefore the defect term still vanishes and we just need to compute the cross section, which is obviously the same as that in phase-I so we just get the same result

$$\mathcal{E}_{\text{III}}^{\text{bulk}}(R_N : R_D) = \frac{c}{2} \log \frac{2x_1'}{\epsilon} . \tag{126}$$

**Phase-IV.** In this phase the cross sections intersect with the brane at two points $M'$ and $N'$ as shown in fig.13(b). By the symmetry with respect to the $x' = 0$ plane, the locations of $M'$ and

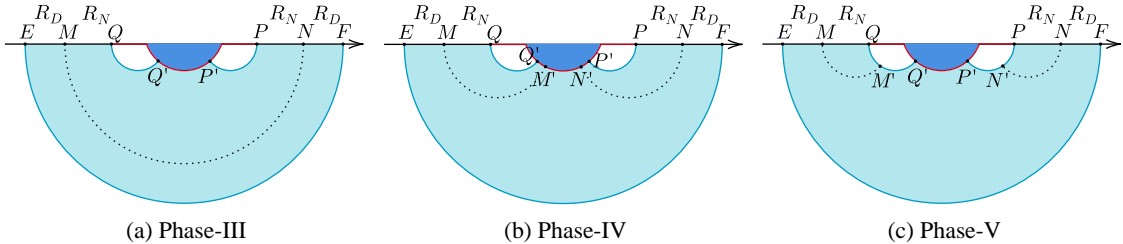

(a) Phase-III  (b) Phase-IV  (c) Phase-V

Figure 13: Possible configurations of the disconnected phase. (a) The cross section dose not terminate on the extremal surface $\widehat{QQ'} \cup \widehat{P'P}$ or on the brane. (b)The cross sections terminate on the brane. (c) The cross sections terminate on the extremal surface $\widehat{QQ'} \cup \widehat{P'P}$.

$N'$ can be denoted as $(\tau'_{1'}, \mp x'_{1'}, z'_{1'})$, or $(-z_{1'} \tan\theta, \mp x_{1'}, z_{1'})$ in the coordinate system $(\tau, x, z)$. The calculation of the extremal surface is similar to eq.(3.7) in [47], which gives

$$
\begin{aligned}
\widehat{MM'}/l &= \widehat{NN'}/l \\
&= \log \frac{(\tau_1 + z_{1'} \tan\theta_0)^2 + (x_1 - x_{1'})^2 + z_{1'}^2}{\sqrt{(\tau_1 + z_{1'} \tan\theta_0)^2 + (x_1 - x_{1'})^2}} \\
&\quad + \operatorname{arctanh} \frac{(\tau_1 + z_{1'} \tan\theta_0)^2 + (x_1 - x_{1'})^2 - z_{1'}^2}{(\tau_1 + z_{1'} \tan\theta_0)^2 + (x_1 - x_{1'})^2 + z_{1'}^2} - \log \frac{4\epsilon}{(\tau'_1 + 1)^2 + x_1'^2}.
\end{aligned}
\tag{127}
$$

The defect contribution from the negativity between $Q'M' \cup P'N'$ and $M'N'$ is given by

$$
\begin{aligned}
&\mathcal{E}_{\text{defect}}(Q'M' \cup P'N' : M'N') \\
&= \lim_{n \to 1} \log \Omega_{Q'}^{2h_n} \Omega_{M'}^{2h'_n} \Omega_{N'}^{2h'_n} \Omega_{P'}^{2h_n} \langle \mathcal{T}_n(Q') \bar{\mathcal{T}}_n^2(M') \mathcal{T}_n^2(N') \bar{\mathcal{T}}_n(P') \rangle_{\text{flat}} \\
&= \lim_{n \to 1} \log \Omega_{M'}^{2h'_n} \Omega_{N'}^{2h'_n} \langle \mathcal{T}'_n(Q') \bar{\mathcal{T}}'_n(Q'') \bar{\mathcal{T}}_n'^2(M') \mathcal{T}_n'^2(M'') \mathcal{T}_n'^2(N') \bar{\mathcal{T}}_n'^2(N'') \bar{\mathcal{T}}'_n(P') \mathcal{T}'_n(P'') \rangle \\
&= \lim_{n \to 1} \log \Omega_{M'}^{2h'_n} \Omega_{N'}^{2h'_n} \langle \mathcal{T}'_n(Q') \bar{\mathcal{T}}'_n(Q'') \rangle \langle \bar{\mathcal{T}}_n'^2(M') \mathcal{T}_n'^2(M'') \rangle \langle \mathcal{T}_n'^2(N') \bar{\mathcal{T}}_n'^2(N'') \rangle \langle \bar{\mathcal{T}}'_n(P') \mathcal{T}'_n(P'') \rangle \\
&= \lim_{n \to 1} \log \Omega_{M'}^{2h'_n} \Omega_{N'}^{2h'_n} \langle \bar{\mathcal{T}}_n^2(M') \rangle_{\text{flat}} \langle \mathcal{T}_n^2(N') \rangle_{\text{flat}}.
\end{aligned}
\tag{128}
$$

In the third line, we use doubling trick. In the fourth line, the eight-point correlator is factorized into contractions assuming the large $c$ limit. In the last line we reverse the doubling trick. Using (41) we get the defect term

$$
\mathcal{E}_{\text{defect}}(Q'M' \cup P'N' : M'N') = \frac{c}{2} \log \frac{2l}{\epsilon_y \cos\theta_0}.
\tag{129}
$$

Add the area term and the defect term, do extremization and we find the extremal solution is at $(-z_{1'} \tan\theta_0, \mp x_{1'}, z_{1'}) = (-\tau_1 \sin\theta, \mp x_1, \tau_1 \cos\theta_0)$. Finally the Rad-Rad entanglement negativity in phase-IV turns out to be

$$
\mathcal{E}_{\text{IV}}^{\text{bulk}}(R_N : R_D) = \frac{c}{2} \left[ \log \frac{x_1'^2 + \tau_1'^2 - 1}{\epsilon} + \operatorname{arctanh}(\sin\theta_0) + \log \frac{2l}{\epsilon_y \cos\theta_0} \right].
\tag{130}
$$

**Phase-V.** In phase-V there is no defect term. As illustrated in fig.13(c), $E_W(R_N : R_D)$ is the geodesic connecting $N(\tau'_1, x'_1)$ and the extremal surface $\widehat{PP'}$, the length of which is given in appendix D and denoted by $L_5$

$$
L_5 = l \log \frac{2\sqrt{\left[ (\tau'_0 - \tau'_1)^2 + (x'_0 - x'_1)^2 \right]\left[ (\tau'_0 \tau'_1 - 1)^2 + (x'_0 x'_1 - 1)^2 + \tau_0'^2 x_1'^2 + \tau_1'^2 x_0'^2 - 1 \right]}}{\epsilon(-1 + \tau_0'^2 + x_0'^2)}.
\tag{131}
$$

Then the Rad-Rad entanglement negativity in phase-V is given by

$$
\begin{aligned}
&\mathcal{E}_{\text{V}}^{\text{bulk}}(R_N : R_D) \\
&= \frac{c}{2} \log \frac{2\sqrt{\left[ (\tau'_0 - \tau'_1)^2 + (x'_0 - x'_1)^2 \right]\left[ (\tau'_0 \tau'_1 - 1)^2 + (x'_0 x'_1 - 1)^2 + \tau_0'^2 x_1'^2 + \tau_1'^2 x_0'^2 - 1 \right]}}{\epsilon(-1 + \tau_0'^2 + x_0'^2)}.
\end{aligned}
\tag{132}
$$

### 7.4.2 Boundary description

**Phase-I.** For phase-I shown in fig.12(a), the island is an empty set so we only need to compute the negativity between two adjacent intervals on a flat CFT, which can be calculated by

$$
\begin{aligned}
\mathcal{E}_{\mathrm{I}}^{\mathrm{bdy}}(R_N : R_D) &= \mathcal{E}_{\mathrm{eff}}(R_N : R_D) \\
&= \lim_{n \to 1} \log \left\langle \mathcal{T}_n(E) \bar{\mathcal{T}}_n^2(M) \mathcal{T}_n(Q) \bar{\mathcal{T}}_n(P) \mathcal{T}_n^2(N) \bar{\mathcal{T}}_n(F) \right\rangle \\
&= \lim_{n \to 1} \log \left\langle \mathcal{T}_n(E) \bar{\mathcal{T}}_n(F) \right\rangle \left\langle \bar{\mathcal{T}}_n^2(M) \mathcal{T}_n^2(N) \right\rangle \left\langle \mathcal{T}_n(Q) \bar{\mathcal{T}}_n(P) \right\rangle,
\end{aligned}
\tag{133}
$$

where in the second line we assume large $c$ limit and factorizes the correlator [39].[14] Using (44) one gets

$$
\mathcal{E}_{\mathrm{I}}^{\mathrm{bdy}}(R_N : R_D) = \frac{c}{2} \log \frac{2x_1'}{\epsilon},
\tag{134}
$$

which agrees with (123) exactly.

**Phase-II.** In phase-II the island is an empty set so we only need to compute the matter term

$$
\begin{aligned}
\mathcal{E}_{\mathrm{II}}^{\mathrm{bdy}}(R_N : R_D) &= \mathcal{E}_{\mathrm{eff}}(R_N : R_D) \\
&= \lim_{n \to 1} \log \left\langle \mathcal{T}_n(E) \bar{\mathcal{T}}_n^2(M) \mathcal{T}_n(Q) \bar{\mathcal{T}}_n(P) \mathcal{T}_n^2(N) \bar{\mathcal{T}}_n(F) \right\rangle \\
&= \lim_{n \to 1} \log \left\langle \mathcal{T}_n(E) \bar{\mathcal{T}}_n(F) \right\rangle \left\langle \bar{\mathcal{T}}_n^2(M) \mathcal{T}_n(Q) \bar{\mathcal{T}}_n(P) \mathcal{T}_n^2(N) \right\rangle.
\end{aligned}
\tag{135}
$$

The four-point function is given by (51)[15]

$$
\lim_{n \to 1} \left\langle \bar{\mathcal{T}}_n^2(M) \mathcal{T}_n(Q) \bar{\mathcal{T}}_n(P) \mathcal{T}_n^2(N) \right\rangle = 2^{c/2} |NP|^{c/2} |NQ|^{c/2} |PQ|^{-c/2} \epsilon^{-c/2},
\tag{136}
$$

thus we have

$$
\mathcal{E}_{\mathrm{II}}^{\mathrm{bdy}}(R_N : R_D) = \frac{c}{2} \log \frac{\sqrt{\left[ (\tau_0' - \tau_1')^2 + (x_0' - x_1')^2 \right] \left[ (\tau_0' - \tau_1')^2 + (-x_0' - x_1')^2 \right]}}{\epsilon x_0'},
\tag{137}
$$

which exactly equals (125).

**Phase-III.** In phase-III there is no area term therefore the entanglement negativity is just given by the matter term

$$
\begin{aligned}
\mathcal{E}_{\mathrm{III}}^{\mathrm{bdy}}(R_N : R_D) &= \mathcal{E}_{\mathrm{eff}}(R_N : R_D) \\
&= \lim_{n \to 1} \log \Omega_{Q'}^{2h_n} \Omega_{P'}^{2h_n} \left\langle \mathcal{T}_n(E) \bar{\mathcal{T}}_n^2(M) \mathcal{T}_n(Q) \bar{\mathcal{T}}_n(Q') \mathcal{T}_n(P') \bar{\mathcal{T}}_n(P) \mathcal{T}_n^2(N) \bar{\mathcal{T}}_n(F) \right\rangle \\
&= \lim_{n \to 1} \log \left\langle \mathcal{T}_n(E) \bar{\mathcal{T}}_n(F) \right\rangle \left\langle \bar{\mathcal{T}}_n^2(M) \mathcal{T}_n^2(N) \right\rangle \left\langle \mathcal{T}_n(Q) \bar{\mathcal{T}}_n(Q') \right\rangle \left\langle \mathcal{T}_n(P') \bar{\mathcal{T}}_n(P) \right\rangle \\
&= \frac{c}{2} \log \frac{2x_1'}{\epsilon},
\end{aligned}
\tag{138}
$$

which is the same as (126).

---

[14]The contractions can be justified by the holographic extremal surfaces shown in fig.12(a).

[15]Note that (51) is correlator of chiral operators thus in (136) all powers double.

**Phase-IV.** The matter term of phase-IV is given by

$$
\begin{aligned}
&\mathcal{E}_{\text{eff}}(R_N : R_D) \\
&= \lim_{n \to 1} \log \Omega_{Q'}^{2h_n} \Omega_{M'}^{2h'_n} \Omega_{N'}^{2h'_n} \Omega_{P'}^{2h_n} \left\langle \mathcal{T}_n(E) \bar{\mathcal{T}}_n^2(M) \mathcal{T}_n(Q) \bar{\mathcal{T}}_n(Q') \mathcal{T}_n^2(M') \bar{\mathcal{T}}_n^2(N') \mathcal{T}_n(P') \bar{\mathcal{T}}_n(P) \mathcal{T}_n^2(N) \bar{\mathcal{T}}_n(F) \right\rangle \\
&= \lim_{n \to 1} \log \Omega_{M'}^{2h'_n} \Omega_{N'}^{2h'_n} \left\langle \mathcal{T}_n(E) \bar{\mathcal{T}}_n(F) \right\rangle \left\langle \bar{\mathcal{T}}_n^2(M) \mathcal{T}_n^2(M') \right\rangle \left\langle \mathcal{T}_n(Q) \bar{\mathcal{T}}_n(Q') \right\rangle \left\langle \bar{\mathcal{T}}_n^2(N') \mathcal{T}_n^2(N) \right\rangle \left\langle \mathcal{T}_n(P') \bar{\mathcal{T}}_n(P) \right\rangle \\
&= \lim_{n \to 1} \log \Omega_{M'}^{2h'_n} \Omega_{N'}^{2h'_n} \left\langle \bar{\mathcal{T}}_n^2(M) \mathcal{T}_n^2(M') \right\rangle \left\langle \bar{\mathcal{T}}_n^2(N') \mathcal{T}_n^2(N) \right\rangle .
\end{aligned}
\tag{139}
$$

Assuming that in the $(\tau, x, z)$ coordinates we have $M'(-y_{M'} \sin \theta, x_{M'}, y_{M'} \sin \theta)$ and $N'(-y_{N'} \sin \theta, x_{N'}, y_{N'} \sin \theta)$, then two-point correlators reads

$$
\begin{aligned}
\lim_{n \to 1} \log \Omega_{M'}^{2h'_n} \Omega_{N'}^{2h'_n} &\left\langle \bar{\mathcal{T}}_n^2(M) \mathcal{T}_n^2(M') \right\rangle \left\langle \bar{\mathcal{T}}_n^2(N') \mathcal{T}_n^2(N) \right\rangle \\
&= \frac{c}{4} \log \frac{l}{y_{M'} \cos \theta_0} + \frac{c}{4} \log \frac{l}{y_{N'} \cos \theta_0} \\
&\quad + \frac{c}{4} \left[ \log \frac{\sqrt{(\tau_1 + y_{M'})^2 + (-x_1 - x_{M'})^2}^2}{\epsilon_y} - \log \frac{4\epsilon}{(\tau_1' + 1)^2 + x_1'^2} \right] \\
&\quad + \frac{c}{4} \left[ \log \frac{\sqrt{(\tau_1 + y_{N'})^2 + (x_1 - x_{N'})^2}^2}{\epsilon_y} - \log \frac{4\epsilon}{(\tau_1' + 1)^2 + x_1'^2} \right].
\end{aligned}
\tag{140}
$$

Extremizing (140) over $y_{M'}$, $x_{M'}$, $y_{N'}$ and $x_{N'}$ gives

$$
y_{M'} = \tau_1, \qquad x_{M'} = -x_1, \qquad y_{N'} = \tau_1, \qquad x_{N'} = x_1.
\tag{141}
$$

Add the area term and finally we obtain the Rad-Rad entanglement negativity

$$
\mathcal{E}_{\text{IV}}^{\text{bdy}}(R_N : R_D) = \frac{c}{2} \left[ \log \frac{x_1'^2 + \tau_1'^2 - 1}{\epsilon} + \operatorname{arctanh}(\sin \theta_0) + \log \frac{2l}{\epsilon_y \cos \theta_0} \right],
\tag{142}
$$

which agrees with (130) precisely.

**Phase-V.** As illustrated in fig.13(c), there is no island cross section on the brane, so the Rad-Rad entanglement negativity in the boundary description reduces to $\mathcal{E}_{\text{eff}}(R_N : R_D \cup I_D)$, which is given by

$$
\begin{aligned}
\mathcal{E}_{\text{eff}}(R_N : R_D \cup I_D) &= \lim_{n \to 1} \log \Omega_{Q'}^{2h_n} \Omega_{P'}^{2h_n} \left\langle \mathcal{T}_n(E) \bar{\mathcal{T}}_n^2(M) \mathcal{T}_n(Q) \mathcal{T}_n(Q') \bar{\mathcal{T}}_n(P') \bar{\mathcal{T}}_n(P) \mathcal{T}_n^2(N) \bar{\mathcal{T}}_n(F) \right\rangle \\
&= \lim_{n \to 1} \log \left\langle \mathcal{T}_n(E) \bar{\mathcal{T}}_n(F) \right\rangle \left\langle \bar{\mathcal{T}}_n^2(M) \mathcal{T}_n(Q) \mathcal{T}_n(Q') \right\rangle \left\langle \bar{\mathcal{T}}_n(P') \bar{\mathcal{T}}_n(P) \mathcal{T}_n^2(N) \right\rangle .
\end{aligned}
\tag{143}
$$

By employing (17) one gets

$$
\left\langle \bar{\mathcal{T}}_n^2(M) \mathcal{T}_n(Q) \mathcal{T}_n(Q') \right\rangle = 2^{c/4} |MQ|^{-2h'_n} |MQ'|^{-2h'_n} |QQ'|^{-4h_n + 2h'_n} \tilde{\epsilon}^{4h'_n},
\tag{144}
$$

where $\tilde{\epsilon}$ is the UV cut-off of $(\tau_1, x_1, 0)$ in $(\tau, x, z)$ coordinates. We also have

$$
\begin{aligned}
|MQ| &= \sqrt{(\tau_0 - \tau_1)^2 + (x_0 - x_1)^2}, \\
|MQ'| &= \sqrt{(\tau_0 + \tau_1)^2 + (x_0 - x_1)^2}, \\
|QQ'| &= 2\tau_0.
\end{aligned}
\tag{145}
$$

One can compute $\left\langle \bar{\mathcal{T}}_n(P') \bar{\mathcal{T}}_n(P) \mathcal{T}_n^2(N) \right\rangle$ using the same method. Insert (144), (145) into (143), we get

$$
\mathcal{E}_{\text{V}}^{\text{bdy}}(R_N : R_D) = \frac{c}{2} \log \frac{\sqrt{[(\tau_0 - \tau_1)^2 + (x_0 - x_1)^2][(\tau_0 + \tau_1)^2 + (x_0 - x_1)^2]}}{\tilde{\epsilon} \tau_0},
\tag{146}
$$

which agrees with (132) after transforming to the $(\tau', x', z')$ coordinates.

### 7.4.3 Time evolution

Here we summarize our results by transforming to the Rindler coordinate via (122) and we get the time evolution. Note that we also assume that $X_0 < X_1 < X_2$.

**Phase-I.**

$$\mathcal{E}_{\text{I}}(R_N : R_D) = \frac{c}{2} \log \frac{2e^{X_1} \cosh T}{\epsilon} \,. \tag{147}$$

**Phase-II.**

$$\mathcal{E}_{\text{II}}(R_N : R_D) = \frac{c}{2} \log \frac{(e^{X_1} - e^{X_0})\sqrt{e^{2X_0} + e^{2X_1} + 2e^{X_0 + X_1} \cosh 2T}}{\epsilon e^{X_0} \cosh T} \,. \tag{148}$$

**Phase-III.** (Note that phase-III is the same as phase-I)

$$\mathcal{E}_{\text{III}}(R_N : R_D) = \frac{c}{2} \log \frac{2e^{X_1} \cosh T}{\epsilon} \,. \tag{149}$$

**Phase-IV.**

$$\mathcal{E}_{\text{IV}}(R_N : R_D) = \frac{c}{2} \left[ \log \frac{e^{2X_1} - 1}{\epsilon} + \text{arctanh}(\sin \theta) + \log \frac{2l}{\epsilon_y \cos \theta} \right] \,. \tag{150}$$

**Phase-V.**

$$\mathcal{E}_{\text{V}}(R_N : R_D) = \frac{c}{2} \log \frac{2(e^{X_1} - e^{X_0})(e^{X_0 + X_1} - 1)}{\epsilon(e^{2X_0} - 1)} \,. \tag{151}$$

The entanglement negativity between nearby radiation and distant radiation experiences two phases. The early time phase corresponds to the connected extremal surface, where the entanglement negativity is given by the minimum value in $\mathcal{E}_{\text{I}}$ and $\mathcal{E}_{\text{II}}$. The late time phase corresponds to the disconnected extremal surface and the negativity is the minimum value in $\mathcal{E}_{\text{III}}$, $\mathcal{E}_{\text{IV}}$ and $\mathcal{E}_{\text{V}}$. In summary, the entanglement negativity between nearby radiation and distant radiation is given by

$$\mathcal{E}(R_N : R_D) = \begin{cases} \min\{\mathcal{E}_{\text{I}}, \mathcal{E}_{\text{II}}\} \,, & T < T_P \,, \\ \min\{\mathcal{E}_{\text{III}}, \mathcal{E}_{\text{IV}}, \mathcal{E}_{\text{V}}\} \,, & T > T_P \,. \end{cases} \tag{152}$$

The result under specific parameters is plotted in fig.14. One can see that before Page time, phase-I and phase-II dominant. At the beginning after Page time, phase-III dominants, which gives the same result as phase-I and later it shifts to phase-V which is a constant all the time.

## 8 Conclusions and discussions

In this paper we have studied entanglement negativity for evaporating black hole based on the holographic model with defect brane. We start from the holographic dual of entanglement negativity for adjacent intervals in $\text{AdS}_3/\text{CFT}_2$ and generalize it to the island formula. To test this formula, we work in $\text{AdS}_3$ with an EOW brane as a bulk defect. Including the contribution from the defect theory on the brane, we propose the defect extremal surface formula for negativity. On the other hand, this model is tightly related to a lower dimensional gravity system glued to a quantum bath. In fact there is a concrete procedure, including both Randall-Sundrum and Maldacena duality, to give a lower dimensional effective description for the same system. We demonstrate the equivalence between defect extremal surface formula and island formula for negativity in $\text{AdS}_3/\text{BCFT}_2$. Extending the study to the model of eternal

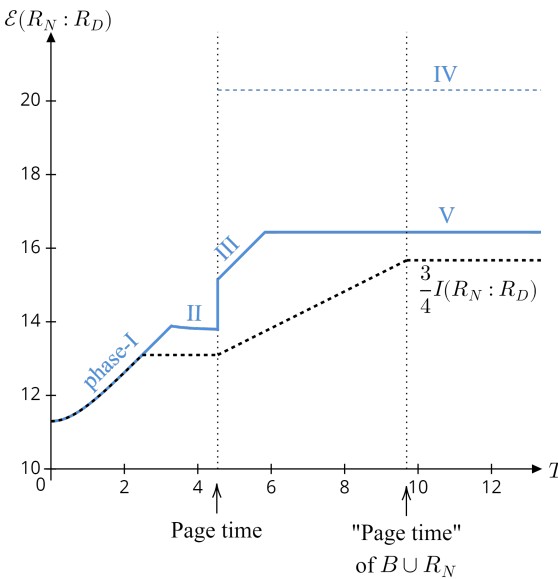

Figure 14: Time evolution of the entanglement negativity between nearby radiation and distant radiation (in the unit of $\frac{c}{2}$) with respect to time $T$. We set $X_0 = 1$, $X_1 = 6$, $l = 1$, $\theta = \frac{\pi}{6}$, $\epsilon = 0.01$, and $\epsilon_y = 0.1$ but note that in such a setting the final result does not rely on $\epsilon_y$. The black dash line shows the evolution of $\frac{3}{4}I(R_N : R_D)$.

black hole plus CFT bath, we find that left black hole-left radiation negativity is always a constant, black hole-black hole negativity decreases until vanishing, radiation-radiation negativity increases and then saturates at a time later than Page time. In all the time dependent cases, defect extremal surface formula agrees with island formula.

Note that there is an alternative holographic proposal for entanglement negativity in adjacent intervals, given by mutual information multiplied by 3/4. One might naively promote 3/4 mutual information to its island formula, which is nothing but a linear combination of three island formulas for von Neumann entropy. We plotted the curves computed from the naively promoted formula, for black hole-black hole negativity and radiation-radiation negativity. We admit that so far we can not prove either one, but just remark that for general quantum system (coupled to gravity), mutual information also measures classical correlation, which is quite different from negativity. So it is unlikely that the island of mutual information can be the final formula. We leave further study on these two different proposals to future.

There are a few future questions listed in order: First, a general holographic dual for negativity. In this paper we mainly focus on the adjacent intervals and the precise holographic dual for negativity of two disjoint intervals is still open. As pointed out by Dong, Qi and Walter [54], there could be dominate contributions from replica non-symmetric saddles. How to find the general holographic dual is essentially the key to find a general island formula. Second, a CFT calculation to justify replica non-symmetric saddles. Inspired from [54], for $n = 2m$ replicas one can take $Z_m$ quotient and there are eight-point functions left over for two disjoint intervals. Developing the CFT techniques to incorporate the replica non-symmetric saddles is crucial to understand negativity in QFT. See also [57] for related discussion. Last, compare our negativity curve to those in other models. For instance, there are recent studies of negativity in JT+EOW models of evaporating black holes [58]. It would be interesting to compare these results. We leave these to future work.

## Acknowledgments

We are grateful for the useful discussions with Tianyi Li, Jinwei Chu and our group members in Fudan University.

**Funding information**   This work is supported by NSFC grant 12375063. YZ is also supported by NSFC 12247103 through Peng Huanwu Center for Fundamental Theory. This work is sponsored by Natural Science Foundation of Shanghai (21ZR1409800) as well as Shanghai Talent Development Fund.

## A   The OPE coefficient

We fix the OPE coefficient by comparing the order $1/2$ Rényi reflected entropy and the entanglement negativity between two adjacent intervals $A = [-\ell_1, 0]$ and $B = [0, \ell_2]$. The Rényi reflected entropy is given by

$$
\begin{aligned}
S_R^{(1/2)}/2 &= \lim_{m\to1}\lim_{n\to1/2}\log\Big\langle\sigma_{g_A}(-\ell_1)\sigma_{g_A^{-1}g_B}(0)\sigma_{g_B^{-1}}(\ell_2)\Big\rangle \\
&= \lim_{m\to1}\lim_{n\to1/2}\log\frac{C_{n,m}}{\ell_1^{4h_n}\ell_2^{4h_n}(\ell_1+\ell_2)^{4nh_m-4h_n}}\,,
\end{aligned}
\tag{A.1}
$$

with [76]

$$
C_{n,m} = (2m)^{-4h_n}\,,\quad h_n = \frac{c}{24}\left(n-\frac{1}{n}\right).
\tag{A.2}
$$

The entanglement negativity between $A$ and $B$ is (17)

$$
\mathcal{E} = \lim_{n_e\to1}\log\Big\langle\mathcal{T}_{n_e}(-\ell_1)\bar{\mathcal{T}}_{n_e}^2(0)\mathcal{T}_{n_e}(\ell_2)\Big\rangle = \lim_{n_e\to1}\log\frac{C_{\mathcal{T}_{n_e}\bar{\mathcal{T}}_{n_e}^2\mathcal{T}_{n_e}}}{\ell_1^{2h'_{n_e}}\ell_2^{2h'_{n_e}}(\ell_1+\ell_2)^{4h_{n_e}-2h'_{n_e}}}\,.
\tag{A.3}
$$

Comparing (A.1) and (A.3), we end up with

$$
\lim_{n_e\to1}C_{\mathcal{T}_{n_e}\bar{\mathcal{T}}_{n_e}^2\mathcal{T}_{n_e}} = \lim_{m\to1}\lim_{n\to1/2}C_{n,m} = 2^{c/4}\,.
\tag{A.4}
$$

## B   Four-point (six-point) function at large $c$ limit

In this appendix we give a calculation of the four-point function in (51) at large $c$ limit using the method in [39,51]. Instead of four-point function, here we first consider a six-point function as illustrated in fig.15, which gives the entanglement negativity between $A$ (the red intervals in fig.15) and $B$ (blue). These six points are parameterized to

$$
x_1 = -a-r\,,\qquad x_2 = -a+r\,,\qquad x_3 = -b\,,\qquad x_4 = b\,,\qquad x_5 = a-r\,,\qquad x_6 = a+r\,.
\tag{B.1}
$$

In the limit $r = \epsilon \to 0$ the six-point function reduces to the four-point function in (51) with UV cut-off $\epsilon$. After a conformal transformation, we can map $\{x_1, x_3, x_4\}$ to $\{z_1, z_3, z_4\} = \{0, 1, \infty\}$ while sending

$$
x_{2,5,6} \to z_{2,5,6} = \frac{(x_{2,5,6}-x_1)(x_4-x_3)}{(x_1-x_3)(x_{2,5,6}-x_4)}\,,
\tag{B.2}
$$

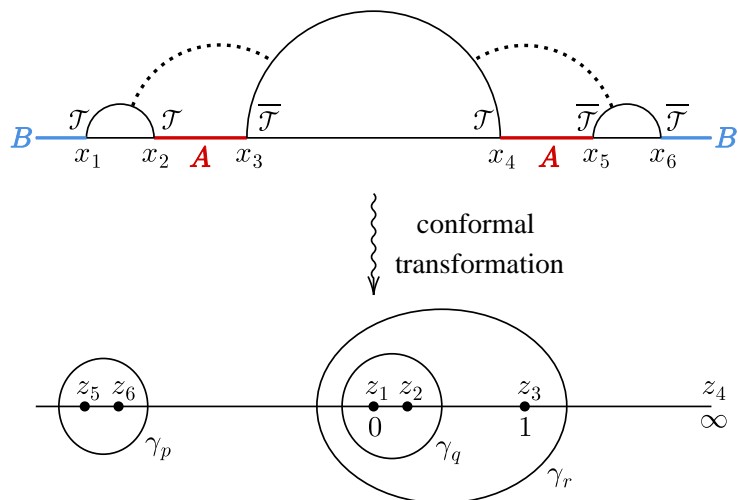

Figure 15: Six point function (B.3) and the holographic dual of its OPE channel. Three points $\{x_1, x_3, x_4\}$ are mapped to $\{z_1, z_3, z_4\} = \{0, 1, \infty\}$ by a conformal transformation. $\gamma_{p,q,r}$ are contours chosen to determine the monodromies (B.9).

as shown in fig.15. At the large $c$ limit with $h_{i=1,\dots,6}/c$ and $h_{a=p,q,r}/c$ fixed,[16] the six point function

$$\left\langle \mathcal{T}(z_1)\mathcal{T}(z_2)\bar{\mathcal{T}}(z_3)\mathcal{T}(z_4)\bar{\mathcal{T}}(z_5)\bar{\mathcal{T}}(z_6) \right\rangle, \tag{B.3}$$

can be approximated by

$$\left\langle \mathcal{T}(z_1)\mathcal{T}(z_2)\bar{\mathcal{T}}(z_3)\mathcal{T}(z_4)\bar{\mathcal{T}}(z_5)\bar{\mathcal{T}}(z_6) \right\rangle \approx c_{12}^p c_{p3}^q c_{56}^r c_{q4}^r \mathcal{F}(z_i)\mathcal{F}(\bar{z}_i), \tag{B.4}$$

where $p, q, r$ label the leading order primaries in the OPE as shown in fig.16, $c_{ij}^k$ are the OPE coefficients, and $\mathcal{F}$ is the six-point Virasoro block, which exponentiates at large $c$ limit [39]

$$\mathcal{F} \sim \exp\left[-\frac{c}{6}f\left(\frac{h_a}{c}, \frac{h_i}{c}, z_i\right)\right], \tag{B.5}$$

with $f$ the semiclassical block which can be computed by solving a monodromy problem. Consider the following differential equation

$$\psi''(z) + T(z)\psi(z) = 0, \tag{B.6}$$

where $T(z)$ is given by

$$T(z) = \sum_{i=1}^{6}\left(\frac{6h_i/c}{(z-z_i)^2} - \frac{c_i}{z-z_i}\right), \tag{B.7}$$

with $c_i$ the accessory parameters satisfying

$$\sum_{i=1}^{6}c_i = 0, \qquad \sum_{i=1}^{6}\left(c_i z_i - \frac{6h_i}{c}\right) = 0, \qquad \sum_{i=1}^{6}\left(c_i z_i^2 - \frac{12h_i}{c}z_i\right) = 0, \tag{B.8}$$

which guarantee that $T(z)$ vanishes as $z^{-4}$ at infinity. The differential equation (B.6) has two solutions $\psi_1$ and $\psi_2$. If we take these solutions on a closed contour around some singular

---

[16]Here $h_i$ denotes the conformal weight of the external operator and $h_a$ the weight of the internal operator. In this appendix we simply take the limit $n_e \to 1$ since we are going to calculate entanglement negativity ultimately, thus we have $h_i = 0$.

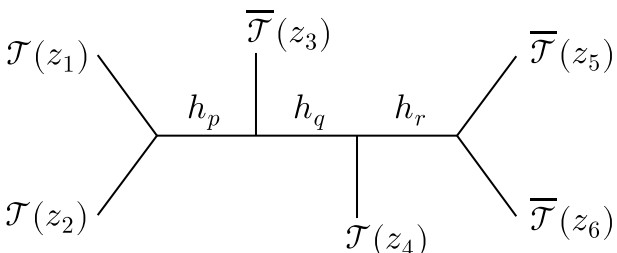

Figure 16: Dominant fusion channel of the six-point function (B.3).

point, as illustrated in fig.15, they will undergo some monodromy

$$\begin{pmatrix} \psi_1 \\ \psi_2 \end{pmatrix} \to M \begin{pmatrix} \psi_1 \\ \psi_2 \end{pmatrix}. \tag{B.9}$$

The accessory parameters $c_i$ can be determined by (B.8) and the following three equations

$$\mathrm{Tr} M_a = -2 \cos \left( \pi \sqrt{1 - \frac{24}{c} h_a} \right), \quad a = p, q, r, \tag{B.10}$$

where $M_{a=p}$ denotes the $2 \times 2$ monodromy matrix for the cycle $\gamma_p$ enclosing $z_1$ and $z_2$ as shown in fig.15, and the same is true for $a = q, r$; $h_p$ is the conformal dimension of the leading operator in the OPE contraction of $\mathcal{T}(z_1)$ and $\mathcal{T}(z_2)$ as shown in fig.16, same for $a = q, r$. From (16) we know that $h_p = h_r = h'_{n=1} = -c/8$ and we also have $h_q = h_{n=1} = 0$ [53].

Now we can solve $c_i$, and they are the partial derivative of $f$ with respect to $z_i$, i.e., $\partial f / \partial z_i = c_i$. Therefore, we can calculate the partial derivative of $\mathcal{E}$ with respect to the coordinate parameters $y = a, b, r$ in (B.1). From (B.4) we obtain

$$\frac{\partial \mathcal{E}}{\partial y} = -\frac{c}{3} \sum_{i=1}^{6} \frac{\partial f}{\partial z_i} \frac{\partial z_i}{\partial y} = -\frac{c}{3} \sum_{i=1}^{6} c_i \frac{\partial z_i}{\partial y}. \tag{B.11}$$

On the other hand, the entanglement wedge cross section (dashed arc in fig.15) is given by (see appendix A of [80])

$$E_W = \frac{\mathrm{Area}[\Gamma]}{4G_N} = \frac{c}{6} \times 2 \log \sqrt{\frac{(a-r)(a+r) - b^2 + \sqrt{((a-r)^2 - b^2)((a+r)^2 - b^2)}}{(a-r)(a+r) - b^2 - \sqrt{((a-r)^2 - b^2)((a+r)^2 - b^2)}}}. \tag{B.12}$$

In the limit $r = \epsilon \to 0$, we have

$$\frac{3}{2} E_W = \frac{c}{2} \log \frac{a^2 - b^2}{b\epsilon}, \tag{B.13}$$

which is the result in (51).[17] The partial derivatives of $\mathcal{E}$ with respect to $a, b, r$ are numerically plotted in fig.17 - 19 and compared with that of $3E_W/2$. The two results match well.

## C The length formula of $L_2$

The computation of $L_2$ in the coordinates $(\tau', x', z')$ is as follow. First we derive a general expression of the length of geodesic shown in fig.20.

---

[17]The difference of a factor two is because (51) is the result after doubling trick.

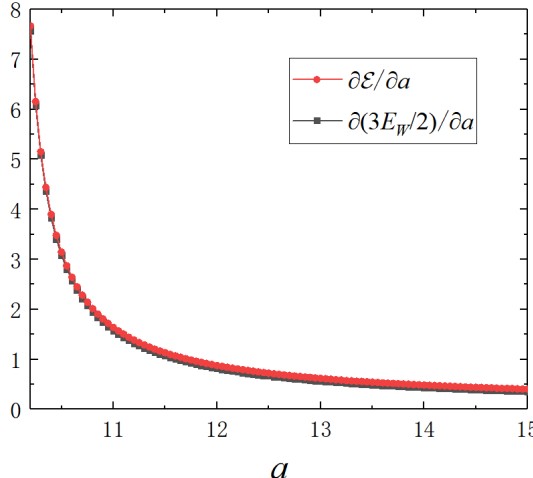

Figure 17: The partial derivatives of $\mathcal{E}$ and $3E_W/2$ (divided by $c/3$) with respect to $a$. We take $b = 10$, $r = 0.01$ and $a$ ranging from 10.2 to 15, in which region the OPE channel dominates.

The radius of the geodesic $R_g$ can be determined by $|AC| = |BC|$

$$|CD| = \frac{d^2 - h^2}{2d}, \quad R_g = |CA| = |CB| = \frac{d^2 + h^2}{2d}. \tag{C.1}$$

According to the metric given by (93), we can compute the length of geodesic

$$L_0(d,h) = \int_{\phi = \phi_\epsilon}^{\phi_B} l\frac{R_g\,d\phi}{R_g\sin\phi} = \left[ l\log\tan\frac{\phi}{2} \right]_{\phi = \phi_\epsilon}^{\phi_B}, \tag{C.2}$$

where $\phi_B$ is the angle between $CA$ and $CB$, $\phi_\epsilon$ comes from the the UV cut-off $z' = \epsilon$ of the asymptotic boundary. Using $(d, h, \epsilon)$ to express $(\phi_\epsilon, \phi_B)$ and we get

$$L_0(d,h) = l\log\frac{d^2 + h^2}{\epsilon h}. \tag{C.3}$$

Then we calculate $L_2$. As shown in fig.12(b), $N'$ can move on the extremal surface so we introduce $\alpha$ to parameterize the position of $N'$. The coordinate of $N'$ can be represented by

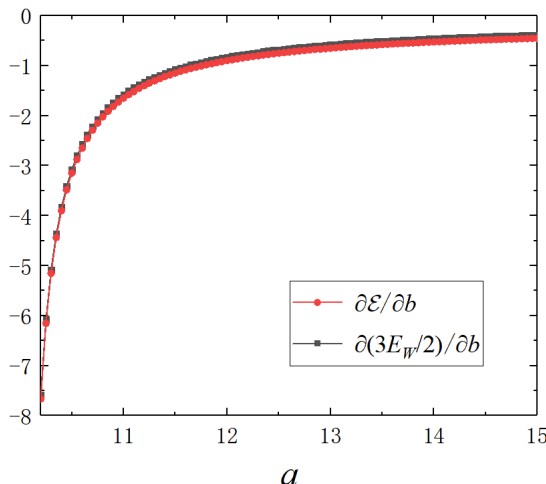

Figure 18: The partial derivatives of $\mathcal{E}$ and $3E_W/2$ (divided by $c/3$) with respect to $b$.

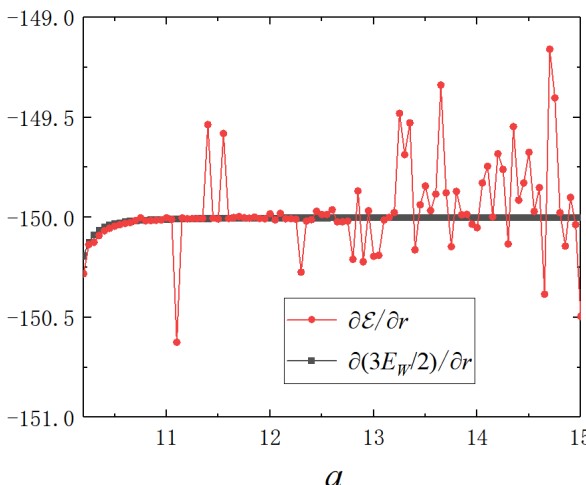

Figure 19: The partial derivatives of $\mathcal{E}$ and $3E_W/2$ (divided by $c/3$) with respect to $r$.

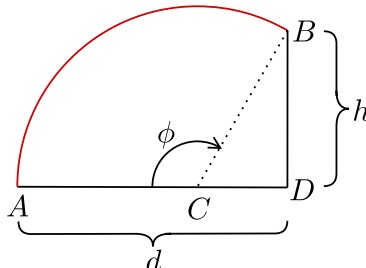

Figure 20: A slice perpendicular to the $\tau'-x'$ plane. The solid red line is the geodesic with $C$ its center. $A$ and $B$ are the ending points of the geodesic, with $A$ on the $\tau'-x'$ plane and $h$ the $z'$ coordinate of $B$. $d$ is the distance between the projections of $A$ and $B$ on $\tau'-x'$ plane.

$(\tau', x', z') = (\tau_0', x_0' \sin\alpha, x_0' \cos\alpha)$ with $\alpha \in (-\pi, \pi)$. So the length of geodesic connecting $N(\tau_1', x_1', 0)$ and $N'$ is given by

$$L_0\left(\sqrt{(\tau_0' - \tau_1')^2 + (x_0' \sin\alpha - x_1')^2}, x_0' \cos\alpha\right) = l \log \frac{(\tau_0' - \tau_1')^2 + (x_0' \sin\alpha - x_1')^2 + x_0'^2 \cos^2\alpha}{\epsilon x_0' \cos\alpha},$$
(C.4)

which is a function of $\alpha$. By extremizing (C.4) with respect to $\alpha$, we get (124) with $\sin\alpha = \frac{2x_0' x_1'}{(\tau_0' - \tau_1')^2 + x_0'^2 + x_1'^2}$.

## D The length formula of $L_5$

We calculate $L_5$ (i.e. the length of the red curve $\widehat{NN'}$ in fig.21) in the coordinates $(\tau, x, z)$. Since the expression of the matric is the same in $(\tau, x, z)$ and $(\tau', x', z')$, we can use (C.3) to write the the length of $\widehat{NN'}$

$$L_0\left(\sqrt{(\tau_0 \sin\gamma - \tau_1)^2 + (x_0 - x_1)^2}, \tau_0 \cos\gamma\right) = l \log \frac{(\tau_0 \sin\gamma - \tau_1)^2 + (x_0 - x_1)^2 + \tau_0^2 \cos^2\gamma}{\tilde{\epsilon} \tau_0 \cos\gamma},$$
(D.1)

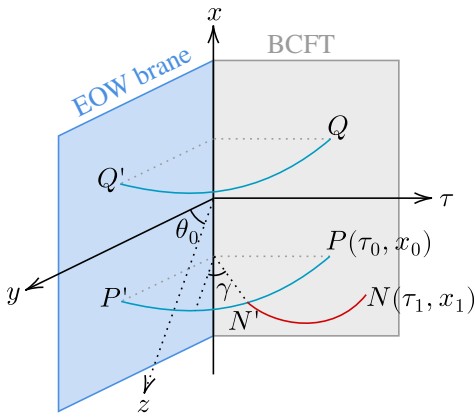

Figure 21: This figure is fig.13(c) viewed in $(\tau, x, z)$ coordinates. The RT surfaces (blue solid lines) are arcs centered on the $x$-axis. The entanglement wedge cross section is drawn with a solid red line.

where $\gamma$ is introduced to parameterize the position of $N'$, $\tilde{\epsilon}$ is the position-dependent cut-off in $(\tau, x, z)$ coordinates which is given by $\tilde{\epsilon} = \frac{4\epsilon}{(\tau_1'+1)^2+x_1'^2}$. Extremizing (D.1) leads to the extremal solution

$$L_5 = l \log \frac{\sqrt{\left[\tau_0^2 + \tau_1^2 + (x_0 - x_1)^2\right]^2 - 4\tau_0^2 \tau_1^2}}{\tilde{\epsilon}\tau_0} . \tag{D.2}$$

We return to $(\tau', x', z')$ by coordinate transformation (90)

$$L_5 = l \log \frac{2\sqrt{\left[(\tau_0' - \tau_1')^2 + (x_0' - x_1')^2\right]\left[(\tau_0'\tau_1' - 1)^2 + (x_0'x_1' - 1)^2 + \tau_0'^2 x_1'^2 + \tau_1'^2 x_0'^2 - 1\right]}}{\epsilon(-1 + \tau_0'^2 + x_0'^2)} . \tag{D.3}$$

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
