# Peer review of "Entanglement Negativity and Defect Extremal Surface"

_SciPost Physics Core, doi:SciPost Phys. Core 7, 027 (2024)_

## Round 1 · Referee Report · Anonymous (Referee 1) · 2023-3-1

Strengths

  1. Paper is well written

  2. Problem is relevant

Weaknesses

  1. Several issues require to be addressed

  2. Originality is low

Report

The authors in their submission have built upon a series of work exploring the doubly holographic prescription for entanglement measures in the framework of AdS/BCFT correspondence with a defect conformal matter on the EOW brane. In particular, the authors have proposed a doubly holographic formula termed as the defect extremal surface (DES) formula for the entanglement negativity (EN) for the corresponding island formula for this mixed state entanglement measure proposed earlier in the literature. Their DES formula is motivated by an earlier holographic proposal relating the EN to the R\'enyi reflected entropy of order $1/2$ which is proportional to the EWCS for certain cases.

Using their DES formula, the authors have obtained the EN for various configurations in holographic BCFT model with bulk defect. They subsequently analyse time dependent situations where they obtain $2d$ eternal black hole on the EOW brane. They obtain the EN for bipartite configurations involving black hole and radiation subsystems. For all the cases considered, they show equivalence between the DES and the island formula for the EN.

While the paper is overall well structured, there are several serious issues listed in the report with implicit assumptions in their proposal which require to be addressed. In my opinion the paper does not meet the acceptance criteria of SciPost Physics and I would recommend the Journal SciPost Physics Core after incorporation of the suggested clarifications and revisions.

Requested changes

In the proposed DES formula in eq. (58), "the defect term" computes the EN between the bulk matter fields where the bipartition is created by the EWCS. For the case considered in this article, the bulk matter fields are only present on the EOW brane. It seems that for any other division of the entanglement wedge of $A \cup B$, as long as the dividing line lands at the same point on the EOW brane, the computations and subsequently the results will remain the same. This restricts the generality and uniqueness of their proposal and should be explained.

In section 3.1, the authors have represented eq. (24) in a way that seems to imply its validity for any general situation which is misleading. In [52,53], the authors state that this equation (specially the prefactor 3/2) is only true for the case where the bulk geometry is 3-dimensional. Given this, the prefactor 3/2 of the area term in the island formula in eq. (28) (defined in the effective $2d$ description) comes under question. This issue needs clarification.

In a recent article (arXiv:2302.10208), the R\'enyi reflected entropy has been shown to not be a correlation measure in the range of the R\'enyi index $n \in (0,2)$ as it does not satisfy the monotonicity condition in this range. In contrast, in [5], EN was shown to be an entanglement monotone. In light of this recent progress, the validity of eq. (24) does not stand on a strong footing which is one of the most crucial assumptions of this article. The authors should add a discussion regarding this.

In section 4, to obtain the 2d-gravity description from the 3d bulk geometry with defect on the EOW brane, the authors have employed a partial RS reduction along with the AdS/CFT duality. But usually in RS reduction of Karch-Randall braneworld models, an inherent CFT appears in the gravity region coupled to the same CFT in the flat bath. However, in this article, it seems that this inherent CFT is absent here and the bulk defect (put in by hand in the action) instead plays a similar role. This issue needs to be explained.

Apart from the above conceptual issues with the proposal, there are several other assumptions made in the computations which are not usually observed in holographic BCFTs and are not explained in the article. Some of them are listed below.

In eq. (50), the authors have utilized the doubling trick to convert a 2-point-correlator in a BCFT to a 3-point-correlator in a chiral CFT. Doubling trick in conventional situations, as the name suggests, doubles the number of operators in a correlation function while mapping it to a chiral CFT. That is not the case here and no explanation is provided for such a deviation. This kind of mapping have been utilized at other instances in the article as well (e.g. eqs. (135,142)) which needs to be clarified.

In section 7.3.2 for the connected phase, the effective entanglement negativity is to be computed between the adjacent intervals $AQ$ and $QO'$ for which the correlation function should be $\langle \mathcal{T}_n(A) \bar{\mathcal{T}}_n^2(Q) \mathcal{T}_n(O')\rangle$ in the BCFT. The authors have ignored the twist operator at point $O'$ due to some unknown reason. I expect the presence of a twist operator at $O'$ in eq. (113) which was present for a similar computation in eq. (102). The authors should provide explanation about its omission.

In eq. (131), the authors have factorized a 6-point correlation function in a CFT to obtain a product of correlation functions which involve 7 twist field operators. In conventional CFTs such an introduction of new operators while factorization is not observed. The authors state this factorization (and the introduction of an extra operator) as a matter of fact without any explanation.

Similar to the previous points, in eqs. (134,135), the authors have converted a 4-point twist correlator in a CFT to a 3-point twist correlator through the utilization of doubling and inverse doubling trick. Again, such an omission of operators from a correlation function is not observed in CFTs. Further justification is required to defend their factorizations.

In the appendix A the authors perform an analysis to obtain the $n\to 1$ limit of the three-point OPE coefficient in eq. (16). I find that a similar analysis for a finite replica index fails to produce a definite answer for the OPE coefficient. Furthermore, following a naive extension of this procedure to the case of the OPE coefficient for the corresponding three-point function for the reflected entropy leads to inconsistencies when compared with appendix C of arXiv 1905.00577. These issues raise serious doubts about the analysis in appendix A.

The first two points mentioned in the beginning leads to ambiguity and uncertainty about the matching of the DES and the island results under this construction which needs to be addressed and clarified. Also the holographic computations presented by the authors in the majority of the article (except section 7.4) follow closely with those in reference [48]. This raises a question about the originality of the work. For these reasons the article does not satisfy the high standards required for publication in SciPost Physics. I would suggest that the work be considered for SciPost Physics Core after the necessary clarifications and revisions.

---

## Round 2 · Referee Report · Anonymous (Referee 1) · 2024-4-21

Report

The authors have addressed all the issues raised in the earlier report satisfactorily. I recommend publication in SciPost Physics Core Journal.

Recommendation

Accept in alternative Journal (see Report)

---

## Round 2 · Author Response

We would like to thank the referee for helpful and constructive suggestions. We have implemented all five points the referee recommends for improvement. More specifically,

(1) We agree to clarify the relation between our set up and the inherent CFT perspective. To our best knowledge, the inherent CFT comes from the old idea that the AdS bulk bounded by Karch-Randall brane (as well as the asymptotic boundary) is expected to be dual to two CFTs and the one on the brane may be called as an inherent CFT. This is not our perspective in this paper. By partial reduction we perform explicit dimension reduction (which may not be applicable in higher dimensions) for the AdS$_3$ gravity action between Karch-Randall brane and the tensionless brane. The resulting 2d gravity is therefore equivalent to 3d gravity in that region, which means that we do not need additional duality to translate this part AdS gravity to some inherent CFT (also to avoid double-counting). In our set up we treat the brane CFT as a bulk defect representing some bulk degrees of freedom from the beginning. Our perspective has received a bunch of tests, such as refs.[46, 47, 48]. We took the referee's suggestion and added some detail discussion on the distinction between our set up and the inherent CFT perspective at the end of Section 4.

(2) We agree to add the justification why the factorizations work. Specifically, for eq.(50) in v2, we first use doubling trick to convert 2-point BCFT correlator to 4-point chiral CFT correlator. Then we calculate the 4-point function assuming large $c$ limit and vacuum block dominance, following the approach in [arXiv:1303.6955] by Hartman. We numerically checked that the dominate channels are indeed the ones corresponding to the holographic configurations. The details of the calculation are included in the new Appendix B. The result of this 4-point function is given in eq.(51) of the current version and one can see that it coincides with 3-point function. The factorizations can also be justified in the same way in other places, such as eq.(135) and eq.(142) in v2.

(3) We agree to include a twist operator insertion at the endpoint of the left subsystem and the referee is right. We corrected our calculation in section 7.3 and updated our results in the current version.

(4) We agree to remove the inverse doubling trick around eq.(134, 135) of v2 and the referee is correct. For the justification of the factorization, precisely the same check can be done by assuming large $c$ and vacuum block dominance as we did in response (2). We include the details of the calculation in the new Appendix B.

(5) We agree to make it clear how the OPE coefficient is obtained. Our OPE coefficient was obtained by matching the half of $n=1/2$ R\'enyi reflected entropy and the entanglement negativity. However we stress that the equality between half of $n=1/2$ R\'enyi reflected entropy and the entanglement negativity is an assumption. The details are included in Appendix A of the current version. The final OPE coefficient is given by taking $m = 1,\ n = 1/2$ for eq.(4.37) of Dutta and Faulkner [arXiv:1905.00577]. We also add footnote 2 to emphasize that the relation between half of $n=1/2$ Renyi reflected entropy and the entanglement negativity is an assumption.

We also corrected many English as well as a few typos of the previous version. We hope the improved version met the clarification the referee suggested and became clearer and more precise.

---

## Round 2 · List of Changes

(1) Discussion about our set up and inherent CFT perspective was added in the end of Section 4 (Page 11).

(2) Justification of why the factorizations work was added after Figure 4 (Page 13). The new Appendix B was added to include the numerical check for such factorization.

(3) The referee's suggestion was taken and accordingly the corrections were made for Section 7.3 (Page 25). Other relevant places were also corrected accordingly.

(4) The previous word "inverse doubling trick" was corrected (eq.(136), Page 30). The justification of factorization follows the same way in response (2).

(5) A derivation of OPE coefficient was added in Appendix A and footnote 2 (Page 6) was added to emphasize that the relation between half of $n=1/2$ Renyi reflected entropy and the entanglement negativity is still an assumption.

---

## Editorial Decision

published